# Immunology Meets Bioengineering: Improving the Effectiveness of Glioblastoma Immunotherapy

**DOI:** 10.3390/cancers14153698

**Published:** 2022-07-29

**Authors:** Zahra Fekrirad, Amir Barzegar Behrooz, Shokoofeh Ghaemi, Arezou Khosrojerdi, Atefeh Zarepour, Ali Zarrabi, Ehsan Arefian, Saeid Ghavami

**Affiliations:** 1Department of Biology, Faculty of Basic Sciences, Shahed University, Tehran 18735-136, Iran; z.fekri@shahed.ac.ir; 2Brain Cancer Research Group, Department of Cancer, Asu Vanda Gene Industrial Research Company, Tehran 1533666398, Iran; am.barzegar.behrooz@gmail.com; 3Department of Microbiology, School of Biology, College of Science, University of Tehran, Tehran 14155-6619, Iran; sokoofeh.ghaemi@gmail.com; 4Cellular and Molecular Research Center, Birjand University of Medical Sciences, Birjand 9717853577, Iran; arezou.khosrojerdi@modares.ac.ir; 5Department of Immunology, Faculty of Medical Sciences, Tarbiat Modares University, Tehran 14115-111, Iran; 6Department of Biomedical Engineering, Faculty of Engineering and Natural Sciences, Istinye University, Istanbul 34396, Turkey; atefeh.zarepour@gmail.com; 7Pediatric Cell and Gene Therapy Research Center, Gene, Cell & Tissue Research Institute, Tehran University of Medical Sciences, Tehran 14155-6559, Iran; 8Faculty of Medicine in Zabrze, University of Technology in Katowice, Academia of Silesia, 41-800 Zabrze, Poland; 9Research Institute of Oncology and Hematology, Cancer Care Manitoba-University of Manitoba, Winnipeg, MB R3E 3P5, Canada; 10Biology of Breathing Theme, Children Hospital Research Institute of Manitoba, University of Manitoba, Winnipeg, MB R3E 3P5, Canada; 11Department of Human Anatomy and Cell Science, University of Manitoba College of Medicine, Winnipeg, MB R3E 3P5, Canada

**Keywords:** glioblastoma, immunotherapy, oncolytic virotherapy, nanomedicine, nanoparticles, autophagy

## Abstract

**Simple Summary:**

Glioblastoma (GBM) is the main malignant brain tumor in adults. It has a poor survival rate and a short disease-free life in spite of all available treatments. The science of cancer immunotherapy is constantly adapting to address the particular needs and problems of a wide range of cancers, including GBM. As of yet, most immunotherapeutic approaches have been unsuccessful, and conventional treatments have been of little effect. Genetically engineered oncolytic viruses (OV) with immunomodulatory transgene expression are now a research focus in the treatment of GBM. There are various challenges that must be addressed in order for immunotherapies to be effective, including physical limitations to medication delivery (e.g., BBB). The present research suggests that nanomedicine and immunotherapy may be a step toward personalized medicine for GBM.

**Abstract:**

Glioblastoma (GBM) therapy has seen little change over the past two decades. Surgical excision followed by radiation and chemotherapy is the current gold standard treatment. Immunotherapy techniques have recently transformed many cancer treatments, and GBM is now at the forefront of immunotherapy research. GBM immunotherapy prospects are reviewed here, with an emphasis on immune checkpoint inhibitors and oncolytic viruses. Various forms of nanomaterials to enhance immunotherapy effectiveness are also discussed. For GBM treatment and immunotherapy, we outline the specific properties of nanomaterials. In addition, we provide a short overview of several 3D (bio)printing techniques and their applications in stimulating the GBM microenvironment. Lastly, the susceptibility of GBM cancer cells to the various immunotherapy methods will be addressed.

## 1. Introduction

As a primary malignant brain tumor, glioblastoma multiforme (GBM) is one of the most aggressive and common cancers, with a poor prognosis and a median survival time of 10 months for GBM patients [1,2,3,4,5,6]. A few targeted therapies exist, and the current gold standard treatment includes surgery, radiotherapy, and temozolomide (TMZ) chemotherapy. GBM is characterized by a lack of immunogenicity and a highly immunosuppressive tumor microenvironment [2]. Because of the resistance of GBM tumors to conventional therapeutics, there is an urgent need to establish alternative strategies to improve current therapeutic approaches. Drugs affecting the common alterations of GBM cancer cells have been investigated as potential targeted therapies [3]. Novel adjuvant chemotherapeutics could be used in combination with standard care. In addition, newly developed molecularly targeted strategies aimed at the tumor cells and the microenvironment could overcome drug resistance [4].

The efficacy of several immunotherapies is under investigation in ongoing studies, including the use of immune checkpoint inhibitors and oncolytic viruses. Immunotherapy with immune checkpoint inhibitors facilitates an effective antineoplastic immune response. These inhibitors interfere with the co-inhibitory receptors and activated pathways by which tumor cells suppress the T-cell response [5]. Using an oncolytic virus to attack tumor cells without harming healthy cells is a potential new treatment option for cancer [6]. Successful clinical application of immune checkpoint inhibitors and oncolytic viruses against GBM is extremely challenging. This review elaborates on the immune-based therapeutic approaches (immune checkpoint inhibitors and oncolytic viruses) and discusses comparative preclinical data and emerging clinical studies. The obstacles and limitations of each immunotherapeutic approach are discussed in the following sections. In addition, promising methods to overcome potential hurdles and strategies to improve therapeutic efficacy are presented. Moreover, we will evaluate different types of nanomaterials (e.g., lipid-based nanoparticles, exosomes, polymeric nanoparticles, injectable hydrogels, and DNA-based nanocarriers) used to enhance immunotherapy and treatment via combination therapy, in which nanomaterials could serve as delivery carriers for antigenic cargo or as stimulating agents.

## 2. Immune Checkpoint Inhibitors in Cancer Immunotherapy

### 2.1. PD-1/PD-L1

PD-1 is one of the genes involved in apoptosis, and a transmembrane protein receptor has been identified as CD279 [7]. By analyzing PD-1-deficient mice, researchers were able to show that PD-1 is a negative regulator of immunological responses [8]. The two ligands for PD-1 are PD-L1 and PD-L2 (also known as CD274 and CD273) [9]. Immune checkpoint inhibitors, particularly those targeting PD-1 and PD-L1, have demonstrated therapeutic efficacy against solid and hematologic tumors [10]. PD-1 signaling negatively modulates T cell-mediated immune responses, which allows malignancies to escape an antigen-specific T cell response [11]. It also contributes to tumor formation and progression by enhancing tumor cell survival. Inhibitors of PD-1 and PD-L1 disrupt the PD-1 axis, reversing T cell suppression and enhancing endogenous antitumor immunity, allowing patients with a variety of malignancies to experience long-term antitumor responses [12].

In light of this, PD-1 signaling is a promising therapeutic target for cancer immunotherapy. For various cancers, six FDA-approved PD-L1 inhibitors are now available: PD-1 inhibitor antibodies nivolumab, pembrolizumab, cemiplimab, and PD-L1 inhibitor antibodies atezolizumab, avelumab, and durvalumab [13,14].

### 2.2. CTLA-4

Cytotoxic T-Lymphocyte-Associated Antigen 4 (CTLA-4) is known as a negative regulator of naïve T cell activation, as evidenced by observations that antibodies against CTLA-4 can promote T cell activation [15] and that the knockout of the Ctla4 gene in mice [16] as well as in humans [17] stimulates lymphoproliferative autoimmune diseases. Later studies established that naïve T cells do not generate detectable CTLA-4 and that Ctla4 is a transcription factor forkhead box (Foxp3) P3 target gene [18], indicating that it is primarily expressed on regulatory T (Treg) cells. The lineage-specific deletion of mouse Ctla4 in Treg cells alone is enough to largely recapitulate the fatal lymphoproliferative disorders connected to germline mutations in the Ctla4 gene [19], implying that CTLA-4 is principally involved in Treg cell functionality. Furthermore, because Ctla4-deficient Treg cells cannot suppress mouse and human effector T cells [19], CTLA-4 appears to be a cell-intrinsic positive (rather than a negative) regulator of Treg function [20]. These findings offer a theoretical foundation for the development of anti-CTLA-4 antibodies for cancer immunotherapy. James Alison’s team was the first to show that blocking the CTLA-4 signal with a monoclonal antibody (mAb) alone can promote tumor regression in a transplanted mouse tumor model [21]. Research suggests that blocking the CTLA-4 inhibitory signal that leads to the reduction in Treg cells at tumor locations is critical for CTLA-4 mAb-mediated anticancer activity [22].

In 2010, ipilimumab, a CTLA-4-blocking human monoclonal antibody, enhanced overall survival in patients with metastatic melanoma [23]. Ipilimumab was approved in the United States and Europe in 2011 for the treatment of unresectable or metastatic melanoma, making CTLA-4 the first immune checkpoint to be targeted in cancer immunotherapy. Despite its survival advantages in patients with metastatic melanoma, ipilimumab therapy is associated with a number of immune-related adverse events, the majority of which affect the gastrointestinal system [23]. Reduced Treg cells and the increased systemic activation of autoreactive T cells in lymphoid tissue are two causes of the numerous side effects of anti-CTLA-4 medication [24].

### 2.3. CD137 and CD47

T cells, NK cells, and antigen-presenting cells express CD137 (4-1BB), an inducible co-stimulatory receptor, which triggers the activation and proliferation of immune cells. Anti-4-1BB agonists increase immune-mediated antitumor activity alone or with other monoclonal antibodies. CD47 is a molecule that acts through the signal regulatory protein alpha (SIRPα) and is prevalently expressed by healthy cells to avoid inappropriate phagocytosis. Various phagocytic cells, such as macrophages and DCs, express SIRPα to generate a “do not eat me” signal. Overexpression of CD47 by tumor cells causes an inhibitory effect on myeloid cells such as macrophages. The CD47–SIRPα axis blockade leads to antitumor activity by increasing macrophage recruitment [25]. CD47 expression is high in GBM and stem cells and is positively correlated with glioma grade. Targeting the CD47–SIRPα axis could be an anti-tumor strategy to restore immunity functions and improve the prognosis of GBM patients [26].

### 2.4. Limitations in Targeting GBM by Immune Checkpoint Inhibitors 

#### 2.4.1. Immune-Microenvironment in the CNS and Local Delivery of Immunotherapeutics

The majority (80%) of all primary malignant brain tumors in adults are glioblastomas, which are also the most aggressive and prevalent variety of this cancer [27]. Different resistance mechanisms exerted by glioblastoma, including intrinsic resistance due to the location of the tumor and physiological characteristics of the central nervous system (CNS), as well as the tumor-suppressing environment, are important challenges in the immunotherapy of GBM [28]. The blood–brain barrier (BBB) is a network of blood vessels made up of special endothelial cells. These non-fenestrated endothelial cells, which are tightly connected to each other, limit the passive transport of immune cells and drugs from the bloodstream to the brain area [28]. It should be noted, however, that pathological conditions cause the BBB to become more permeable, increasing the probability of immune cells and medicines penetrating the brain [29]. One of the new methods that has recently been considered is low-intensity pulsed ultrasound (LIPU) to increase the permeability of the BBB for the better delivery of chemotherapy drugs and immunotherapy agents [30]. After preclinical studies, the safety and efficacy of this method have been confirmed in phase I/II clinical trials [31,32].

In addition to the physical barrier, the unique combination of the immune cells and the tumor environment in GBM reduces the effectiveness of immunotherapy. One of the most frequent types of immune cells in a healthy brain environment are macrophages, which can be either brain-resident macrophages (microglia) or peripheral monocyte-derived macrophages [33]. In GBM, most cells that make up a tumor mass are macrophages derived from peripheral monocytes, which are called tumor-associated macrophages (TAM) because of their unique properties [34]. These macrophages, which co-express surface markers of both M1 and M2, lack CD40 and CD86 molecules and are not able to activate T lymphocytes [33]. In addition, TAMs have the ability to produce IL-10 and TGF-β and induce Th2, which is reminiscent of M2 [35]. APC s such as TAMs express STAT3 as a transcription factor, which causes the secretion of anti-inflammatory cytokines, and in a positive loop, these cytokines increase STAT3 expression [36]. In this way, TAMs change the environment in favor of the tumor. Additionally, since peripheral myeloid cells are the source of the majority of these M2-like macrophages, treatments that target peripheral myeloid cells before reaching the tumor site may be effective.

Studies have shown that CNS immune surveillance occurs primarily in the meningeal vasculature far from the tight junctions in the endothelium [37]. Accordingly, most of the glioblastoma classes are bereft of tumor-infiltrating lymphocytes [38]. Due to the weak infiltration of T cells into GBM, this tumor is called “cold” [39]. In addition, different studies have revealed that T cells in the GBM microenvironment are usually anergic, ignorant, or exhausted [40]. Despite the fact that systemic lymphopenia is the main factor in lacking an effective immune response in GBM, the poor expression of tumor-specific antigens and the overexpression of non-specific tumor-associated antigens in GBM limits T lymphocyte access to enough antigens [34]. Additionally, GBM causes the anergy of T cells by decreasing the surface expression of co-stimulatory molecules and increasing the expression of inhibitory molecules such as CTLA-4 and PD-L1 [41,42]. Clinical research has demonstrated that tumor growth and aggressiveness are directly correlated with CTLA-4 expression in GBM [41]. According to the research by Ricklefs et al., GBM secretes extravesicles that contain PDL-1, which prevents T lymphocytes from proliferating and becoming activated [43]. The study of cells isolated from the GBM microenvironment showeda high expression of inhibitory molecules. In other words, GBM induces exhausted-T cells by upregulating inhibitory molecules [44]. This could be a cause of the low success rate of a single-blocker ICI treatment in GBM patients. Lower CD4+ T cell numbers [45] and higher CD8+ CD28- T cells (an exhausted phenotype) [46] are linked to poor prognosis and decreased overall survival in glioblastoma cases.

The development of tolerance in T cells that reach the tumor microenvironment presents a further challenge to GBM immunotherapy. By increasing the expression of the apoptosis-inducing molecule FAS Ligand (FASL) at the level of astrocytes [47] and by increasing the number of Treg cells [48], the tumor employs two different strategies to induce tolerance. In a healthy brain, FASL is detected only on the surfaces of neurons, while astrocytes are not [49]. Additionally, Treg cells are absent from the brain under normal conditions, whereas they make up approximately 27% of the population of invading T cells in GBM [48]. Thus, Treg cells inhibit the activity of migrating immune cells to the tumor environment by secreting inhibitory cytokines (TGF, IL-10) and expressing inhibitory molecules (CTLA-4) [50]. As a result, therapies based on the elimination of Treg cells, or the restriction of their function can probably stop tumor cells from inducing tolerance. This issue reveals that glioblastoma occurs, somehow, due to impairment of the immune system, and that stimulation of a T cell response might alter the outcome in the glioblastoma patients.

In addition, cancer stem cells from GBM have been identified as major players in the resistance to chemotherapeutic drugs [51] and tumor recurrence [52]. Hence, efficient therapeutic approaches should be designed to target these cell populations above and beyond the more differentiated cell types within the tumor mass. It needs to be considered, however, that the cranial space in the CNS is inherently intolerant to edema. This issue results in more restrictions to the level of flow and pressures related to the delivery of therapeutics and should be taken into account with regard to possible immunotoxicity due to cytokine release stimulated by immunotherapeutic compounds [53].

#### 2.4.2. The Proposed Solution: The Effector Immune Cells Reaching the Tumor Area

To date, numerous immunotherapeutic studies have been conducted to improve the effectiveness of immune cells in the GBM environment. According to the results of these investigations, polarization induction strategies, the use of genetically engineered cells, and the use of inhibitory antibodies and molecules are all potentially promising treatments for GBM.

Due to the importance and role of TAMs in inducing a tumor suppressant environment, various studies have focused on inhibiting these M2-like macrophages and inducing polarization in these cells. The colony-stimulating factor-1 receptor (CSF-1R) has been the subject of numerous studies that have demonstrated how inhibiting this receptor can interfere with macrophage chemotaxis and activity in GBM and enhance the efficacy of radiation and chemotherapy [54,55,56]. In addition, a published study by Zhang et al. exploited in situ genetically engineered macrophages with mRNA-harboring nanoparticles to be differentiated from an M2 to an M1 phenotype. In this study TAMs received mRNA encoding IRF5 and IKKβ, and after that, the cells reprogrammed and showed the characteristics of M1. This strategy has been shown to promote overall survival in animal models of GBM [57]. Several studies have indicated that STAT3 knockout [58] or the use of miR-124 [59] reduces Treg cells while increasing anti-tumor activity in the GBM animal model. According to the study by Ott and collages, STAT3-blockers were able to stop the progression of the tumor in a GBM animal model [60]. In addition, a phase I clinical trial with the use of STAT3-inhibitors in GBM patients has been reported, but its results have not yet been published (NCT01904123).

As previously indicated, ignorance, anergy, exhaustion, and tolerance often occurs in T cells that successfully infiltrate GBM. Hence, various studies have focused on restoring the activity of these cells in the GBM environment. Immune checkpoints as effective molecules in the dysfunction of invading T cells have been studied in various studies. 

Recently, a phase 1 clinical trial was registered that used radiotherapy, neoantigen vaccination, and Pembrolizumab (PD 1-inhibitor) used simultaneously. However, the results of this trial have not been published (NCT02287428). ICIs such as PD/PDL-1inhibitors have been approved for the treatment of many solid tumors, but different clinical trials on GBM are still ongoing (NCT02311920, NCT02017717, and NCT03233152). A fully human monoclonal antibody (utomilumab) that stimulates 4-1BB is currently being evaluated, either as a monotherapy or in different combinations with mogamulizumab, rituximab, and avelumab, in multiple clinical trials in patients with advanced solid tumors [25]. Furthermore, a phase I trial in 44 patients with recurrent GBM is currently ongoing and is exploring the therapeutic effectiveness of targeting the checkpoint molecules anti-LAG-3 (lymphocyte activation gene) or anti-CD137 alone and in combination with anti-PD-1 (phase I trial, NCT02658981). A recent study showed that a laboratory-developed anti-CD47 antibody inhibited the in vivo growth of U251 cells and promoted the inhibition capacity by Taxol (microtubule-stabilizing drug) [61]. Another study showed that the CD47 blockade alone could not stimulate glioma cell phagocytosis in human and murine, but that the CD47 blockade combined with TMZ significantly impacted pro-phagocytosis in murine and human GBM cells [62]. Thus, targeting more recently identified immune checkpoints such as CD137 and CD47, alone or in combination with other therapeutic approaches to enhance their efficacy, could be of therapeutic value in the treatment of tumors such as GBM.

One of the crucial factors generating tolerance in the GBM environment is the presence of Treg cells. Therefore, many attempts have been made to remove these cells from the GBM environment. Various preclinical studies have shown that targeting the IL-2 receptor (anti-CD25) inhibits the function of Treg cells and increases survival in the GBM-mouse model [63,64]. In addition, the use of daclizumab in combination with temozolomide and the vaccine against EGFRvIII has been shown to selectively eliminate Treg in patients with GBM [65]. 

The results of EGFRvIII-targeted vaccination [66] and CART therapy [67], as a GBM-specific antigen, have shown that targeting this antigen alone not only did not increase patient survival but also GBM downregulated the antigen. In the other clinical trial, CMV antigens were applied to mature DCs in vitro, and subsequently, these adult DCs were given to patients to activate T cells. Patients’ survival rates increased as a result of the findings [68]. 

#### 2.4.3. Immunosuppressive Nature of GBM

Besides the high inhibitory checkpoint expression and an impaired ability of peripheral lymphocytes to transport to the tumor microenvironment, the secretion of immunosuppressive cytokines and chemokines could play a significant role in the immunosuppressive nature of GBM [69]. Intercellular communication of immune cells shapes the immune system and could be facilitated by direct cell contact or the use of soluble mediators such as cytokines and chemokines. These small cell-signaling protein molecules can be upregulated in various and mediated several anti-tumor effects in some cancers [70]. Cytokines and chemokines exhibit immunomodulatory effects and affect the microenvironment of glioblastoma tumors. Recent studies have shown that chemokines expression of CXCL2, CX3CL1, CCL5, and CCL2 and chemokine receptors CCR5, CX3CR1, CXCR2, CXCR4, CXCR6m and CXCR7 are significantly high in glioblastoma tumors [71]. Moreover, high gene expression of the pleiotropic cytokine IL-6 which suppresses T cell functions was associated with poor survival in glioblastoma patients. IL-6 induces the anti-apoptotic pathways in glioblastoma cells and promotes invasion. The hypoxic glioblastoma cells initiate autophagy with the assistance of IL-6 to support tumor cell growth in a nutrient-deprived microenvironment [72]. Another important cytokine that plays an important role in cell growth and differentiation, angiogenesis, and immunoregulation is TGF-β. The endogenous anti-tumor immunity could be defeated by this cytokine. TGF-β also inhibits the release of IFN-γ and TNF-α (pro-inflammatory cytokines) from lymphocytes which leads to a limited response against grade III glioma cells.

Some recent clinical trials are trying to overcome the anti-tumor effects of several cytokines and chemokines in glioblastoma patients. One of the best studied signaling pathways in the GBM microenvironment is the CXCR4/CXCR7/CXCL12 signaling pathway. A CXCL12 inhibitor, NOX-A12, is under evaluation (clinical phase I/II study) in combination with irradiation in newly diagnosed GBM patients (NCT04121455). AMD3100, a CXCR4 inhibitor, could cause a reduction in vessel density and tumor growth in mice and rat GBM models. A phase II clinical study is currently evaluating the efficacy of whole-brain irradiation with TMZ and AMD3100 in newly diagnosed GBM patients. The progression-free survival, overall survival, and the out-of-field occurrence of recurrence will be measured (NCT03746080). In another phase II study is currently evaluating the efficacy, safety, and impact on the tumor microenvironment of the combination of Tocilizumab (antagonist of il-6 receptor), Atezolizumab (anti pd-l1), and radiotherapy in recurrent glioblastoma (NCT04729959). Additionally, the effects of genetically engineered natural killer cells containing deleted TGF-βR2 and NR3C1 on 25 patients with recurrent glioblastoma are under investigation (not yet recruiting) in a phase 1 clinical trial (NCT04991870). 

## 3. Oncolytic Viruses

Cancer therapy involving viruses is considered a promising and novel strategy. Virotherapy can be divided into approaches that use non-replicating viruses (as gene delivery vector systems) or oncolytic replicating viruses [73]. Oncolytic viruses (OVs) stimulate the immune system and trigger apoptosis, particularly in tumor cells. The OV replicates indefinitely inside tumor cells and finally kills them. Additionally, a large number of antigens and signaling molecules are produced as a consequence of the infection, triggering immune responses against the tumor [74]. Because the oncolytic virotherapy uses the amplification of replicative viruses, the viral progeny and the danger-associated molecular patterns (DAMPs) can trigger innate and adaptative immune responses [75]. Viral infection of the tumor induces both an antiviral and an anticancer immune response, which results in tumor elimination by immunogenic cell death (ICD). Additionally, OVs may be genetically modified to be tumor-specific and to reduce their non-specific pathogenicity and/or increase their immunogenicity toward tumors [76]. Various proposed types of OVs that can be used to treat GBM are presented in Figure 1.

### 3.1. Viruses Proposed as Glioma Oncolytic Agents

#### 3.1.1. DNA Viruses

##### Herpes Simplex Virus Type I

Herpes Simplex Virus Type 1 (HSV-1, a member of the *Herpesviridae* family) is a DNA virus with an enclosed double-stranded DNA genome. The virus’s capacity to infect and proliferate in brain tissue makes it an excellent option for glioma therapy. HSV-1 wild type enters cells by attaching its viral glycoprotein D (gD) to the cell surface protein CD111, also known as nectin-1 [77]. Nectin-1 is highly expressed in gliomas compared to normal tissue [78]. In 2021, a modified HSV-1, named DELYTACT (teserpaturev/G47Δ), contains a triple mutation (i.e., deletion of ICP34.5, ICP6, and α47 genes) and was approved in Japan to treat malignant glioma [79]. 

##### Adenovirus

Adenoviruses are icosahedral non-enveloped viruses with a double-stranded DNA genome. A total of 57 serotypes have been described in humans thus far, some of which are pathogens for humans [80]. Conditionally replicative adenoviruses (CRads) have been identified as the best option for use as oncolytic viral treatment. Several versions of CRad have been produced in recent years, with promising outcomes in clinical investigations of GBM [81]. 

##### Vaccinia Virus (VV)

Vaccinia (a member of the Poxviridae family) is a double-stranded DNA virus with an encased genome. Importantly, VV enabled the eradication of smallpox. Because the virus does not need cell surface receptors for entrance and penetrates the host cell through membrane fusion instead, it has the ability to infect practically every mammalian cell type in any tissue. Accordingly, VV is an interesting tool for developing oncolytic viruses against glioma; in addition, it is favorably characterized by a quick and effective non-integrative replication cycle, strong cell-to-cell dissemination capabilities, and its genetic alteration is relatively straightforward [82]. 

##### Myxoma

Myxoma virus (MYXV) is a poxvirus family member with an enclosed double-stranded DNA genome. The reason this virus is a suitable candidate for oncolytic treatment is its capability to infect and propagate in cells without the IFN system [83]. MYXV inhibits the expression of MHCI in infected glioma cells, hence eliminating NK cells. One downside of employing MYXV as an oncolytic virus in vivo is its limited ability to multiply beyond the location of tumor injection [84]. 

##### Parvovirus

Parvovirus belongs to the Parvoviridae family of single-stranded DNA viruses. The capsid structure of these viruses is icosahedral. To date, 134 distinct parvoviruses have been identified, all of which are capable of infecting a variety of animal species. Clinical work on parvovirus H-1 for glioma treatment is outlined in Table 1 [85]. 

#### 3.1.2. RNA Viruses

##### Measles

Measles virus (MV) is an enveloped, negative single-stranded RNA virus belonging to the *Paramyxoviridae* family. The fusion (F) and hemagglutinin (H) proteins of MV are required for its anticancer action in GBM. MV penetrates cells through an interaction between the viral H protein and the CD46 cell receptor. CD46 is a protein found on almost every human cell and is overexpressed in tumor cells [86,87]. 

##### Vesicular Stomatitis Virus (VSV)

VSV is an enveloped, negative single-stranded RNA virus of the *Rhabdoviridae* family. VSV entry is mediated by its glycoprotein spike (G) and a ubiquitous host cell receptor, the low-density lipoprotein receptor (LDL-R); because this is a generally expressed receptor, the virus can enter almost every cell type [88]. VSV preferentially replicates in malignant cells and less so in normal cells [89]. Induced aberrations in the IFN pathway in tumor cells caused by VSV IFN sensitivity particularly highlights the potential of this virus as an oncolytic candidate for glioma. Aside from predominantly targeting malignant cells, VSV recombinants could be generated that could increase tumor susceptibility to chemotherapeutic agents and/or the host immune response [90]. 

##### Reovirus

*Reoviridae* is a family of double-stranded RNA non-enveloped viruses. Reoviruses can overtake and precisely replicate cells activated by the Ras pathway, commonly present in cancer cells such as glioma [91,92].

##### Newcastle Disease Virus (NDV)

NDV is an enveloped, negative single-stranded RNA virus belonging to the *Paramyxoviridae*. NDV mainly infects avian species while having marginal pathogenicity in humans [93]. Following NDV infection, a type I IFN response is stimulated in human cells [94].

##### Seneca Valley Virus Isolate 001 (SVV-001)

Seneca Valley virus isolate 001 (SVV-001) is a non-enveloped, positive single-chain RNA virus and a member of the *Picornaviridae* family. SVV-001 is serologically linked and identical to 12 swine picornaviruses. Therefore, it is thought to be contaminated with porcine trypsin. SVV-001 has been shown to exhibit neuroendocrine tumor specificity and oncolytic activity [95,96].

##### Poliovirus

Poliovirus belongs to the *Picornaviridae* family. These viruses contain positive single-strand RNA and are enclosed with a capsid [97]. Poliovirus can cause neurotoxicity, although this tropism can be eliminated by replacing the internal ribosome entry site (IRES) of the poliovirus vaccine sabin strain with the non-virulent human rhinovirus type 2 (HRV2) [98].

##### Sindbis

Sindbis is a tiny alphavirus with a positive single-stranded RNA genome encapsulated by a *Togaviridae* capsid protein, and birds are its natural hosts [99]. When the virus attaches to the 67-kDa high-affinity laminin receptor (LAMR), which is overexpressed in cancer cells, infection ensues [100].

##### Rift Valley Fever Virus (RVFV)

RFVF is a single-stranded RNA virus belonging to the *Bunyaviridae* family with many known hosts, including several domestic animals and humans [101]. Clinical studies involving Reovirus and their effects on glioma are also summarized in Table 1.

### 3.2. Oncolytic Viruses Expressing Immunomodulatory Transgenes

To break the immunosuppressive barriers in GBM, immunostimulatory therapeutic transgenes have been inserted into oncolytic viruses, and the induced antitumor immune response inhibits the growth or kills tumor cells. The oncolytic-mediated tumor destruction is achieved by activating and stimulating innate and adaptive immune responses. Recent preclinical and clinical trials have studied the safety and efficacy of oncolytic viruses carrying immunomodulatory transgenes. Immune-stimulating genes which have been virally introduced to GBM tumor cells are interleukins (IL), immune checkpoint inhibitors, immune stimulators, tumor necrosis factor (TNF)-related apoptosis-inducing ligand (TRAIL), E-cadherin, FMS-like tyrosine kinase 3 ligand (Flt3L), and tumor suppressors (Figure 2). 

#### 3.2.1. Interleukins

IL-12 regulates innate as well as adaptive immunity and is used to improve the antitumor efficacy of the oncolytic virus against GBM. It augments the IFN-γ production, growth, and cytotoxic activities of NK cells and T lymphocytes and inhibits angiogenesis [102]. Several studies have reported that HSV engineered to express IL-12 inhibited tumor growth and increased survival in a murine model of GBM [103,104,105].

Treatment of U87 cells and human glioma stem cell-derived GBM xenograft models by G47∆-mIL12 (HSV expressing murine IL-12) inhibited GBM angiogenesis and prolonged survival compared to control HSV [104]. Another study showed that the median survival of xenograft GBM models (patient-derived) improved by M002, an IL-12 expressing HSV [106]. M032, a genetically engineered HSV-1 that expresses IL-12, is currently being evaluated in phase I clinical trials involving patients with recurrent/progressive GBM (NCT02062827). 

IL-15 is another crucial factor in immune regulation (e.g., by activating NK and CD8+ cells) and is used in cancer research because of its induction of solid antitumor immune responses [107,108]. For example, adenovirus expressing IL-15 induced innate and adaptive immune mechanisms and inhibited angiogenic activity, leading to antitumor efficacy [109].

#### 3.2.2. TRAIL and Flt3L

Other cytokines such as TRAIL and Flt3L have been inserted into oncolytic HSV as well. Flt3L induces the maturation and proliferation of DCs and NK cells and decreases metastases by hindering tumor progression. The use of HSV expressing Flt3L resulted in the complete eradication of the GBM tumor and an increased life span in animal models [110]. Evaluation of the clinical effects of the combination of direct tumor cell targeting by the thymidine kinase gene and the immune-mediated tumor cell targeting (Flt3L gene) is ongoing (phase I, NCT01811992). TRAIL could cause apoptosis in tumor cells by activating the TNF–CD95L axis. Because of tumor-specific apoptosis induction, TRAIL is considered a therapeutic candidate for GBM treatment [111]. However, TRAIL produces off-target toxicity and has a short half-life [112]; oncolytic viruses could be a promising strategy to limit such systemic off-target toxicity. Tamura et al. designed an oncolytic transgene HSV encoding a secreting TRAIL (G47Δ-TRAIL) that exhibited cytotoxicity against resistant GBM cells (TRAIL or HSV) in comparison with control oncolytic HSV lacking the TRAIL gene. Intra-tumoral inoculation of HSV-TRAIL inhibited tumor invasion and prolonged survival of mice with HSV-resistant intracerebral GBM cells [113]. In another promising study, the oncolytic HSV-TRAIL treatment of mice bearing a recurrent human GSC-derived tumor with TMZ-insensitivity inhibited tumor growth and extended animal survival (40% cure rate). The improved efficacy of this transgene oncolytic virus against chemoresistant tumor cells was associated with strong apoptosis induction [114].

#### 3.2.3. Immune Checkpoint Inhibitors

Immune checkpoint inhibitors support T cell activity by the blockage of negative regulators of T cell function. Because tumors such as GBM are characterized by low T cell infiltration, they are inherently less susceptible to the effects of immune checkpoint inhibitors. One possible solution for this problem is the induction of CD8^+^ T cell responses against tumor cells by oncolytic virotherapy to increase the efficacy of immune checkpoint inhibitors. The inflammatory response to a combination of oncolytic virotherapy along with checkpoint block leads to the expression of PD-1 on T cells and PD-L1 on tumor cells [115]. The therapeutic potency of immune checkpoint inhibitors in GBM animal models and patients with recurrent GBM (anti-PD-L1 antibody) has been demonstrated previously [116,117]. In addition, oncolytic HSV armed with an antibody against PD-1 exhibited a durable antitumor response in GBM mouse models, and surviving mice from the first GBM challenge rejected the second GBM implant [118].

In a recent study, the antitumor effect of an oncolytic virus (Delta-24-ACT) armed with a 4-1BB ligand was evaluated alone and in combination with an immune checkpoint inhibitor (anti-PD-L1 antibody) in preclinical glioma models. Results showed that Delta-24-ACT exerted a cytotoxic effect on human glioma cell lines and tumors in the murine models, implying that the expressed 4-1BBL was able to stimulate T cells in vitro and in vivo. The combination of Delta-24-ACT with an anti-PD-L1 antibody extended the median survival and generated immune memory, which indicated the potential of combining immune-virotherapy with immune checkpoint inhibitors as a promising and effective strategy for the treatment of poorly infiltrated tumors such as glioma [119].

#### 3.2.4. Immune Stimulators

Glucocorticoid-induced TNFR family-related gene (GITR) is upregulated in most immune cell types upon activation. GITRL, the ligand of GITR and a vital member of the tumor necrosis factor receptor superfamily, is expressed in the immune system, especially in antigen-presenting cells. The co-stimulatory signal of GITR to antigen-primed T cells (both CD4^+^ and CD8^+^) leads to proliferation and effector function. In contrast to antibodies, co-stimulatory ligands could be inserted into oncolytic viruses, and the use of these armed viruses against tumor cells would result in the expression of these co-stimulatory molecules and the amplification of antitumor T-cell activity. GITRL-armed Delta-24-RGD increased the number of central memory CD8^+^ T cells and prolonged the survival of glioma-bearing immunocompetent mice. Rechallenge of the surviving mice with a second glioma cell implantation did not lead to tumor growth [120].

The OX40 ligand binds to the unique immune co-stimulator OX40 on T cells and increases their activation. In glioma-bearing mice, an oncolytic adenovirus expressing OX40L (Delta-24-RGDOX) resulted in cancer-specific immunity. Administration of oncolytic adenovirus activated the tumor-specific lymphocytes and increased the proliferation of CD8^+^ T cells specific to tumor-associated antigens. Immune stimulation mediated by the intra-tumoral injection of both Delta-24-RGDOX and anti-PD-L1 antibody synergistically inhibited glioma tumor cells and increased the survival of mice [121].

#### 3.2.5. E-Cadherin

E-cadherin (E-cad) is a calcium-dependent adherent molecule and a ligand for the lymphocyte co-inhibitory receptor KLRG1 (killer cell lectin-like receptor G1) that blocks the lysis of E-cad-expressing cells by NK cells. Overexpression of E-cad on tumor cells infected by engineered HSV expressing CDH1 prolonged the survival in GBM-bearing mouse models. The improved survival was correlated with an enhanced viral spread rather than the inhibition of NK cell activity [122]. 

Genetically engineered oncolytic viruses that express immune stimulators, such as cytokines, could increase the immune response by improving immune cell infiltration and stimulation of subsequent immune network cascades. Overall, the described studies on using modified OVs provide promising therapeutic effects against gliomas. A summary of ongoing trials evaluating the clinical impact of OVs on gliomas is presented in Table 2. OVs that express immunomodulatory transgenes have been used increasingly to treat glioma brain tumors but primarily in preclinical work. Still, as for every (novel) therapeutic approach, there are some limitations to the effectiveness of this strategy.

**Figure 2 cancers-14-03698-f002:**
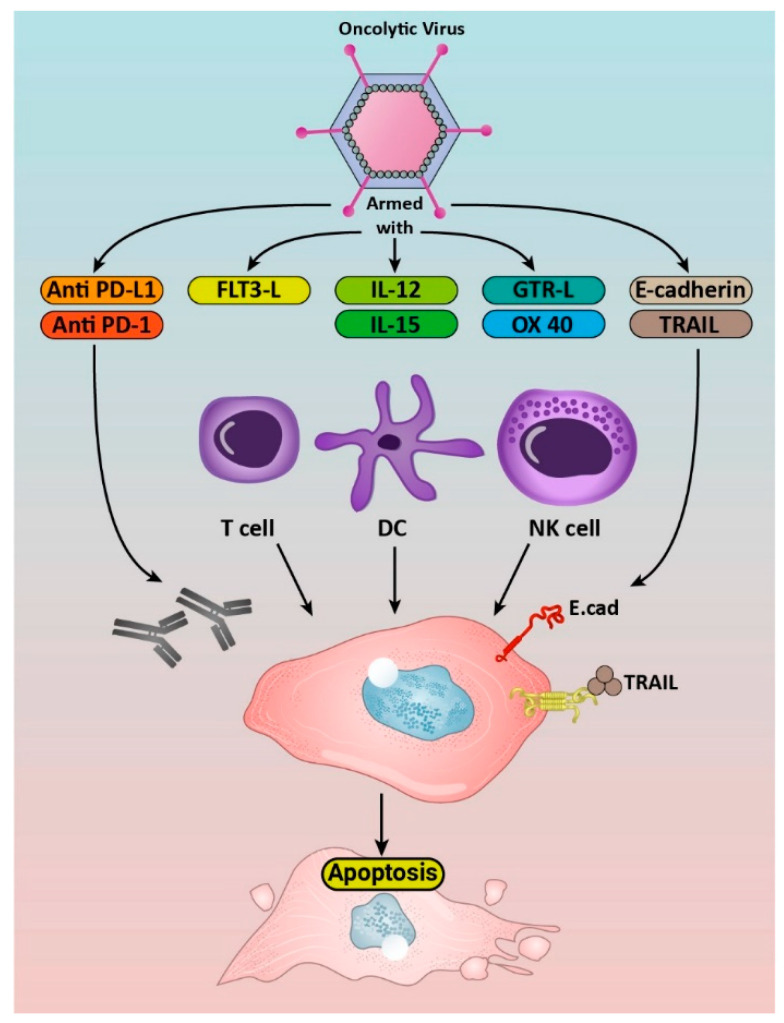
Schematic representation of armed oncolytic viruses against Glioblastoma. OVs express immunomodulatory transgenes: interleukins (IL 12,15), TRAIL and Flt3L. OVs express immune checkpoint inhibitors (anti-PD-L, anti-PD-1). OVs express immune stimulators (GITRL, OX40). OVs express E-cadherin.

### 3.3. Limitations of Targeting GBM by Oncolytic Viruses and Possible Solutions

#### 3.3.1. Innate Antiviral Response

An innate antiviral response, mediated mostly by type-I interferons, serves as the initial line of defense against viral infections. To prevent viral spread, infected cells secrete type-I interferons in response to viral nucleic acids. Interferons released by macrophages promote antigen presentation and NK cell activity as well as the formation of antigen-specific adaptive immune responses. The release of type-I interferons attracts innate immune cells to destroy virus-infected cells. This critical step primes adaptive antitumor and antiviral immunity; however, the premature elimination of virus-infected cells impairs virus spread within the tumor [115].

OVs used in current therapies are either of a natural origin or genetically engineered and are sensitive to the antiviral effect of interferons. The dysfunctional IFN-1 signaling pathways in cancer cells help these OVs to exert their functions. It has been shown that human GBM samples produce type-I interferon and respond to them. Furthermore, the abnormal effects of these interferons on GBM immunosurveillance indicate that a deficient IFN-I system could not be taken for granted in GBM. Therefore, OVs that potently activate the innate immune system and can resist the resulting antiviral response would have the best chance of replicating in GBM tumors [115].

##### Proposed Solution: Genetic Engineering

As indicated, the inclusion of immunoregulatory genes within the genome of OVs can potentiate the adaptive immune response against tumor antigens. In addition to cytokines (such as interleukins), the inserted transgenes could include chemokines, co-stimulatory or inhibitory receptors, immune ligands, and combinations of these genes. Genetic engineering allows OVs to enhance the induction of tumor-specific immunity. One of these newly engineered viruses is T-Vec, a modified HSV-1 virus carrying the human GM-CSF gene to increase the number of DCs and consequently to extend the activity of cytotoxic CD8^+^ lymphocytes [123].

##### Proposed Solution: MicroRNA

MicroRNAs (miRNAs) are small non-coding RNAs expressed to mediate the post-transcriptional regulation of gene expression. miRNA deregulation has been widely associated with many types of cancer. Incorporating miRNA target sequences could control exogenous gene expression and viral tropism in specific tissues and result in enhanced therapeutic effects of transgenes expressed by OVs. Indeed, oncolytic virotherapy in combination with miRNA strategies has been proposed as a potent and safe cancer therapy.

Recently, the oncolytic capacity of a neurovirulent Semliki Forest virus (SFV)-4 in GBM and neuroblastomas was evaluated. The insertion of miRNA target sequences for miR134, miR124, and miR125 into SFV4miRT attenuated neurovirulence in normal mouse CNS cells, while its oncolytic capacity was retained. These results revealed that SFV4miRT could be a candidate for the treatment of GBM (and neuroblastoma) with low IFN-β secretion [124].

Martikainen et al. aimed to increase the therapeutic potency of SFV as it was hindered by IFN-I-mediated antiviral signaling. The efficacy of an IFN-I-resistant SFV construct (SFV-AM6) was evaluated in vitro, ex vivo, and in vivo. Despite strong IFN-I signaling, SFV-AM6 treatment led to the immunogenic apoptosis of GL261 cells. SFV-AM6-124T (detargeted miRNA-124) selectively replicates in glioma cells and, when administered intraperitoneally, can infect orthotopic GL261 gliomas. Combined immunotherapy with SFV-AM6-124T and anti-PD-1 increased immune cell infiltration in GL261. The results indicated that SFV-AM6-124T can overcome obstacles of innate anti-viral signaling, and that combining SFV-AM6-124T with anti-PD1 enhances the inflammatory response in a GL261 glioma model [125]. 

In other recent work, the unwanted toxicities of oncolytic picornavirus (vMC_24_ΔL) were addressed using miRNAs. vMC_24_NC was generated by the insertion of neuron- and cardiac/muscle-specific miRNA targets into the viral genome. The in vivo analysis of the syngeneic murine plasmacytoma model showed excellent therapeutic efficacy without significant virus-mediated toxicity [126]. Combining OVs and miRNAs showed reduced toxicity and enhanced potency in preclinical models. If the application proves safe and effective in clinically relevant animal models and future clinical trials, miRNA-detargeted OVs hold high promise in cancer therapeutics.

#### 3.3.2. Virus Delivery

Upon systemic delivery, before reaching the tumor, OVs have to face anti-viral immune cell responses and neutralization by complement factors and/or antibodies [127]. Furthermore, the physical BBB in the CNS prevents viral particles from reaching the tumor and regulates the vascular to extravascular passage of the virus [128].

##### The Proposed Solution: Intra-Tumoral Administration

Fundamental issues of OV usage in patients remain challenging; these include nonspecific viral toxicity, viral DNA integration or viral latency in the host, attenuated virus/incomplete responses, and complexity or low capacity for genetic modifications. To address some of these challenges, it would help to design and create an infiltrative growth pattern supported by optimal routes for vehicle delivery [129]. Previous studies have indicated the neutralization and elimination of intravascularly injected viruses by antibodies in the host [130,131]. To overcome these problems, the careful orthotopic injection of OVs for malignant gliomas is recommended. However, activated macrophages and microglia may prevent viral expansion by engulfing virus particles [115,132] Taking this into consideration, intra-tumoral administration is the most effective way to control local concentrations. 

##### The Proposed Solution: Promote CNS Tropism

Some OVs show tropism to neuronal tissue or use immune cells as carriers, which enables them to cross the BBB, leading to them infecting and killing tumor cells in the CNS. Mesenchymal and neural stem cells can serve as carriers for OVs because of their tumor tropism. Several preclinical experiments have indicated that stem cell carriers loaded by OVs can be efficiently delivered to GBM tumors when injected systemically [133]. Published results from pre-clinical and clinical trials have confirmed that MSCs are suitable carriers for recombinant viruses because of their tumor-tropic and immune-evasive capabilities [134].

Cellular tropism of viruses, based on cell entry or post-entry characteristics, should also be considered as it might limit the infection of specific cell types in the tumor microenvironment [135,136].

#### 3.3.3. Targeting Autophagy to Enhance Oncolytic Virus-Based Cancer Therapy

Autophagy is an evolutionarily conserved process that supports cellular homeostasis and cell survival under different conditions of stress [137,138,139,140]. This process is a promising but challenging therapeutic target to enhance the efficacy of current cancer therapy. Autophagy is a vital regulator of the host response against viral infections, and the interactions between viruses and autophagic machinery are complex and virus specific. These interactions include: autophagy limits virus replication as an innate immune defense mechanism, viruses disrupt autophagy to facilitate their replication, and viruses utilize autophagy as a scaffold to promote their replication [141].

The induction of autophagy by OVs has been shown to result in a reduction in innate immune responses against viruses leading to extended viral replication. Defective IFN-1 signaling in some cancer cells permits OVs to replicate into malignant cells, whereas in other cancer cells the IFN-1 response is intact and is employed to induce tumor resistance to oncolytic virotherapy [142]. Although viruses affect autophagy in a both virus- and (target) cell-specific manner, OVs can promote autophagy and enhance tumor cell death. In contrast, autophagy nourishes cancer cells metabolically and protects them from OV-mediated lysis and cytotoxicity. Adenovirus enhances its spread among cancer cells by using autophagy. It has been shown that by using rapamycin, an autophagy promoter, the replication of chimeric Ad5/F35 increased. Conversely, 3-methyladenine, an autophagy inhibitor, restricted the production of viral structural proteins and consequently inhibited adenovirus replication [143,144]. In addition, combined treatment with oncolytic adenovirus, TMZ and metronomic cyclophosphamide enhanced tumor cell autophagy and led to antitumor immune responses [145].

Infection of glioblastoma cells with the OBP-405 adenovirus, without autophagy activators, promotes cell lysis without concurrent replication. Combining oncolytic adenovirus with TMZ diminishes tumor growth and stimulates different anti-tumor immune responses, resulting in immunogenic cell death [145]. Autophagy is regulated by numerous upstream molecules such as IRE1α, which participates in endoplasmic reticulum (ER) stress responses. In contrast to previous studies, Li et al. showed that autophagic inhibitors enhance the oncolytic effects of the M1 virus. They found that IRE1α expression is negatively correlated with malignant glioma progression, and altering autophagic signaling could augment oncolytic immunotherapy [146].

Several factors that include cancer cell type and OV variants determine the efficacy of targeting autophagy in oncolytic virotherapy. Both the inhibition and activation of autophagy could increase oncolysis by immunogenic cell death in a cell- and OV-specific manner. Due to its dual function, it is challenging to target autophagy unilaterally for the augmentation of oncolysis in various cancers.

## 4. GBM–Immunotherapy–Bioengineering

### 4.1. Engineering and Biology: Two Pairs of Eyes Are Better Than One

Engineering is not just about engines. In debates about engineering speak in biology, a common picture of what engineering is and how engineers operate recurs. A well-informed engineer at a drawing board creates designs for a polished and optimized product that “merely” has to be created in this picture of engineered artifacts, which are mostly mechanical or electrical gadgets. While this picture may capture some instances of engineering, it does not adequately represent the field’s depth and breadth. It is important to note that it omits important aspects of engineering relevant to biology. A deliberate, goal-directed designer does not create biological systems as he/she does in engineering. However, this does not imply that the processes of evolution and human design are diametrically opposed. When we look at the design processes used by certain human engineers and innovators, we see a lot of parallels to evolution. Trial and error play an important part in the work of even the most imaginative innovators, for example. Human engineers, on the other hand, are more goal-oriented in the near term than nature, but their predictions about what a new invention may be used for often fall flat. As a result of their idealistic ideas of engineering, critics of the biology–engineering nexus typically disregard the iterative and error-prone process that really takes place [147]. One of the most severe kinds of cancer, malignant brain tumors have low survival rates, which have not altered in the previous 60 years. In part, this is due to the particular anatomical, physiological, and immunological barriers that the brain presents. Innovative engineering solutions may be facilitated by the unique interaction of these obstacles. Cancer immunotherapy, which uses the body’s own immune system to fight cancer, is becoming a common treatment for a wide range of tumors. Anatomical, physiological, and immunological challenges arise when working with the brain, which is a crucial organ. There are inherent technical problems in creating medicines that must function inside the brain and brain tumor immune milieu because of these specific anatomical and physiological restrictions. We may be able to use a wider range of technologies to develop personalized therapy techniques that can meet the specific biological restrictions of treating inside the brain, thereby allowing improved brain cancer immunotherapies [148].

### 4.2. Nanomedicine and Glioblastoma Immunotherapy

The introduction of nanoscience and nanotechnology in medicine has led to the emergence of a promising field with significant hopes for the treatment of hard-to-treat diseases such as GBM. Nanomaterials could act as carriers for the targeted delivery of therapeutic and diagnostic agents to the cancer tissue or could themselves be applied as therapeutics. They are promising tools capable of improving the bioavailability of drugs, reducing unwanted toxicity in healthy tissues, delivering drugs to the desired site of action, and releasing drugs in a controllable manner, thus reducing the number of daily doses through extended-release formulations. In addition to these pharmacokinetic and pharmacodynamic benefits, nanomaterials could be on the forefront of treating brain-related diseases as they are capable of non-invasively passing the BBB by surface engineering (Figure 3) and can deliver therapeutics from the nose to the brain via olfactory nerves [149,150,151,152,153]. 

Thus, the application of nanomaterials in immunotherapy could have several advantages, of which the targeted delivery of immunological drugs and enhancing their bioavailability in the target tissue are (clinically) most important. Moreover, based on their design, nanomaterials can exhibit endosomal escape properties, thereby preventing the degradation of immunomodulatory components (such as DNA and RNA). Considering these advantageous nanomaterial characteristics, the combination of nanotechnology and immunotherapy may lead to the introduction of effective (novel) methods for GBM therapy [155,156,157]. 

#### 4.2.1. Different Types of Nanomaterials with Applications in GBM Treatment

NPs are ideal candidates for GBM cancer treatment because of their unique features, including a high surface-to-volume ratio, a narrow size distribution, a high drug loading capacity, and excellent surface modification. They could enhance the therapeutic performance of their cargo molecules via improving bioavailability and enhancing accumulation in the tumor region. Moreover, NPs can be used to deliver several agents simultaneously (combination therapy), thereby improving treatment effectiveness and decreasing the probability of drug resistance. In addition to therapeutic capability, nanomaterials can be utilized as diagnostic tools to detect cancer cells and monitor the treatment process [158,159,160]. In the field of cancer immunotherapy, nanomaterials can be used as safe carriers with the goal of modulating immune responses via transferring and controlling the release of different immune stimulators/suppressors such as antigens [161,162], small interfering RNAs (siRNAs) [163,164], drugs [165,166], immune adjuvants [167,168], and various combinations of these compounds [169,170]. NPs could also enhance the activation of antigen-presenting cells (APC) via antigen delivery, targeting the activation of T cell-specific antigens, and adjust the immunosuppressive nature of the tumor microenvironment (TME) [156].

The main categories of different types of nanomaterials that can be considered for the immunotherapy of glioblastoma are described in the following sections:

##### Lipid-Based Nanoparticles

Lipid-based nanocarriers, a group of nanomaterials that include different types of liposomes, micelles, nanostructured lipid carriers (NLC), and solid lipid nanoparticles, are considered good candidates for drug delivery to GBM. They are biocompatible carriers with the ability to deliver both hydrophobic and hydrophilic therapeutic and targeting agents [171]. Liposomes are spherical vesicles containing bilayers of phospholipids with similarities to the cell membrane, and their structural features can be optimized for specific applications. Surface functionalization of these nanosystems with polymers such as polyethylene glycol (PEG) improves their stability and allows for the attachment of targeting agents on the surface [172]. Liposomes have been used to treat glioblastoma by targeting the immune system [173,174,175]. Tumor-associated macrophages (TAMs) are involved in tumor progression by creating an immunosuppressive tumor microenvironment (TME). Chlorogenic acid is one of the therapeutic agents that could be used as an immunomodulatory agent to promote a transition from the polarizing M2 TAM phenotype to M1. Thus, mannosylated liposomes were used to encapsulate and deliver this therapeutic compound to glioma cells, and the in vitro and in vivo results confirmed the inhibition of tumor cell growth by promoting the conversion of the macrophage phenotype from pro- to anti-tumorigenic (Figure 4). The fabricated nanoformulation showed an anticancer effect via two different methods; on one side, the mannose-targeted carrier could promote the TAM polarization to M1 phenotype via activating the phosphorylation of the nuclear factor kappa-light-chain-enhancer of activated B cells (NF-κB) and suppressing the signal transducer and activator of transcription 6 (STAT6). On the other side, chlorogenic acid could also enhance the suppression of the STAT6 and activation of the STAT1. This polarization was also confirmed via increasing in the expression level of cytokines such as interferon gamma (IFN-γ) and tumor necrosis factor alpha (TNF-α) which led to a significant reduction in the amount of interleukin 10 (IL-10), an immunosuppressive cytokine that promotes tumor growth [176]. In other work, the combination of three different herbal drugs (curcumin, resveratrol, and epicatechin gallate) loaded inside liposomes was shown to exhibit onco-immunotherapeutic effects via triggering the activation of caspase 3 in GBM cells and suppressing GBM stem cells in the tumors. This combination therapy led to the repolarization of M2 macrophages to the tumoricidal M1-phenotype and the recruitment of activated NK cells [177].

Solid lipid nanoparticles (SLNs) and NLCs are two other lipid-based nanocarriers made up of natural lipids with low immunogenicity and high biocompatibility. The surface functionalization of these materials with targeting agents makes them capable of accumulating inside their target sites. Due to the higher capacity of drug loading and drug release in a more controllable manner, NLCs are typically preferred over SLNs [178,179]. Neutral and positively charged lipid nanocapsules containing lauroyl-modified gemcitabine were shown to effectively target myeloid-derived suppressor cells (MDSCs), one of the primary cell types inhibiting the anti-tumor immune response [180]. The combination of radiation therapy and immunotherapy has also been used to reduce the growth of tumor cells and improve survival rate. SLNs were functionalized with a cyclic peptide iRGD (CCRGDKGPDC) to target the delivery of siRNAs and impact both PD-L1 and the epidermal growth factor receptor (EGFR). Radiation therapy enhanced the uptake of NPs by the tumor cells, allowing the siRNAs to effectively act against both targeted genes, that led to suppressing the proliferation and aggressiveness of the tumor cells along with prevention of immune suppression (Figure 5). Utilizing SLNs in here enhanced the stability and bioavailability of siRNAs and prevented their degradation before they reached the targeted site [181]. 

A combination of chemotherapy and immunotherapy for the treatment of GBM has also been studied. Chitosan-coated lipid nanocapsules were functionalized with CpG genetic material as an immune modulator. Paclitaxel (PTX), as a type of anticancer drug, was also encapsulated inside the lipid part of the carrier. The in vivo results showed that the combined use of the therapeutic agents significantly enhanced the survival rate of animals compared to the controls and the group treated with PTX alone [182]. 

##### Exosomes

Exosomes are cell-derived vesicles with the ability to transport a variety of molecules (from proteins and lipids to nucleic acids), while being targeted to specific cells. They are ideal candidates for the encapsulation and delivery of therapeutic agents. Despite differences in their biological functions, exosomes commonly express characteristic proteins, including CD8, CD9, CD63, and proteins used for endocytosis and cargo sorting, such as flotillin and the tumor-susceptibility gene 101 protein [183]. Furthermore, they may act as double-edged immunomodulatory agents leading to either the promotion or suppression of cancer cells [184].

Exosomes derived from tumor cells could represent promising candidates for delivering therapeutic agents to the tumor cells. A type of DC-based vaccine was developed for the immunotherapy of GBM by co-loading tumor-derived exosomes and α-galactosylceramide (α-GalCer) on DCs. As such, the α-GalCer was able to activate the immune system’s invariant NK cell [185]. Importantly, GBM-derived exosomes also contain different components, including (a) melanoma-associated antigen 1 (MAGE-1 protein), a specific antigen for the glioma tumor, (b) major histocompatibility complex class I (MHC-I), an essential factor for antigenic peptide capturing, (c) 70 kilodalton heat shock proteins (Hsp70s), a critical chaperone for binding to the DCs, and (d) intercellular adhesion molecule-1 (ICAM-1), a transmembrane glycoprotein useful for cell targeting and immune response activation. The produced vaccine showed anti-tumor activity via strong activation of the tumor-specific cytotoxic T lymphocytes, modulating the immunosuppressive environment, and suppressing the immune tolerance [186].

Exosomes derived from neural stem cells (NSCs) have been applied for GBM therapy as well, and were used to transport antisense oligonucleotides targeting STAT3 in the glioma microenvironment. The constructed exosomes stimulated human DCs, glioma-associated microglia, and the production of IL-12 along with the induction of nuclear factor κB (NF-κB) signaling. The in vitro and ex vivo results confirmed a higher effectiveness of nucleotide-loaded exosomes than the nucleotide alone for cancer treatment [187]. 

In another study, exosome mimetics were derived from extracted NK cells, and their effects on glioblastoma tumor cells were evaluated. In vitro tests confirmed the cytotoxic effects of these immune-derived vesicles. Moreover, in vivo results demonstrated the antitumor activity of the exosomes indicated by reductions in the size and weight of tumor tissues. Treatment with the exosome mimetics resulted in an increased abundance of apoptosis protein markers (including cytochrome-c, cleaved-caspase 3, and cleaved-poly ADP ribose polymerase (PARP)) concomitant with a decrease in cell survival markers (such as phosphorylated extracellular signal-related kinase (p-ERK) and phosphorylated Protein kinase B (p-AKT)). Furthermore, it was shown that the presence of NKp30 ligands (originating from the NK cells) on the surface of these exosomes was responsible for their accumulation at the tumor site (Figure 6) [188]. 

##### Polymeric Nanoparticles

Polymeric NPs are considered the largest and most prevalent group of nanomaterials used for GBM treatment. In general, polymeric NPs are divided into two main groups: natural polymers, such as chitosan, collagen, cellulose, and dextran, and synthetic polymers such as poly(lactic acid) (PLA), poly(glycolic acid) (PGA), polybutyl cyanoacrylate (PBCA), polyethyleneimine (PEI), polyethylene glycol (PEG), and polyglycerol (PG), in different shapes [189]. Rapamycin inhibits the mammalian target of rapamycin (mTOR) (associated with the regulation of cell growth, proliferation, and immune responses) and is therefore an excellent therapeutic compound for the treatment of GBM. However, its low bioavailability, dose-dependent toxicity, and poor penetration into the brain and tumor restrict its application. In an attempt to address these issues, targeting TAMs with a polymeric dendrimer conjugated to rapamycin was evaluated in a clinically relevant orthotopic syngeneic model of GBM. The fabricated dendrimer—rapamycin localized specifically within TAMs, where it released rapamycin into the tumor microenvironment. This resulted in TAM reprogramming and anti-tumor activity [190].

Chitosan NPs containing siRNA were investigated for the intranasal delivery of galectin-1 (Gal-1), a natural galactose-binding lectin that is overexpressed in GBM tumors. Gal-1 acts as an immune-suppressive agent by inducing apoptosis in activated CD8^+^ T cells, antagonizing T cell signaling, and blocking the secretion of pro-inflammatory cytokines. Treatment with the siRNA-loaded NPs significantly decreased Gal-1 expression, which could improve the immune system’s effectiveness against tumor cells [191]. 

In another study, polycarbonate vesicles were fabricated from the self-assembly of poly (ethylene glycol)-polycarbonate-spermine and functionalized with apolipoprotein E peptide (ApoE) as a targeting agent that can cross the BBB and accumulate inside the glioma. The presence of spermine in the structure of this nanoparticle provides a high capacity for encapsulating CpG ODN, a toll-like receptor 9 agonist that has been used as an immunotherapeutic agent. This was a redox responsive formulation, which led to the stimulation of pro-inflammatory cytokine production and DC maturation in vivo, as well as the reprogramming of the tumor immune microenvironment by inducing more immunogenic cell death (ICD) (of tumor cells), CD8^+^ T cells, and activated antigen presentation cells (APCs). Combined use of this “smart” brain-permeable nano-immunoadjuvant with radiotherapy further boosted the anti-cancer effect and improved the survival rate in mice (Figure 7) [192].

Smart polymeric NPs can also be used as carriers for immune-stimulating/suppressing agents. For instance, anti-PD-L1 (to achieve immune checkpoint blockade (ICB)) was encapsulated inside a polymeric NP via a pH-responsive bond. The polymeric NPs were fabricated from the attachment of choline analog 2-methacryloyloxyethyl phosphorylcholine (MPC), which acted as a targeting agent as well, to short-chained poly (ethylene glycol) methacrylate (PEGMA) via a free-radical polymerization process. The acidic pH of the GBM tumor tissue induced the detachment of anti-PD-L1 from the nanoparticles, allowing anti-PD-L1 to interact with PD-L1 in the cancer cells, which resulted in the suppression of the immune checkpoint and activation of T cells [193].

Polymeric NPs have the potential for broad application in combination therapies for GBM. For instance, a multi-functional NP containing a photosensitizer (5-ALA), a magnetic upconversion NP (MUCNP), two different polymeric compounds (chitosan and poly (γ-glutamic acid), a targeting agent (des-Arg9-Kallidin (d-K)) for crossing the BBB, and an immune checkpoint inhibitor (anti-PD-L1) was developed for the combination of photodynamic- and immunotherapy for primary and metastatic GBM. The transformation of 5-ALA into protoporphyrin IX following 980 nm laser irradiation led to the production of reactive oxygen species (ROS) inside the tumor and improved apoptotic cell death. In addition, this photodynamic therapy promoted the intratumoral infiltration of cytotoxic T lymphocytes. They checked the effect of the presence of the anti-PD-L1 antibody on the activation of the tumor-associated macrophages (MΦ) via an evaluation of the expression level of costimulatory molecules such as CD86, interleukin 12 (IL-12), and tumor necrosis factor-α (TNFα). The immunofluorescence results showed that the expression of these components in cells treated with the antibody alone and the nanoformulation-contained antibody was higher than the control and the nanoformulation without an antibody, that confirmed the effectiveness of utilizing this antibody in the activation of the immune response. Accordingly, the combined use of photodynamic therapy and immunotherapy could be an effective method for the treatment of GBM cancer affecting the immune system and changing the microenvironment of cancer tissue (Figure 8) [194].

##### Injectable Hydrogels 

Hydrogels are biocompatible polymeric components that exist in a gel form and can switch to their functional mode in response to a specific condition to locally deliver therapeutic components. Low invasive administration, easy formulation, the capacity to load different therapeutic agents, and the ability for controlled release make these carriers ideal candidates for various treatment strategies. Regarding immunotherapy, hydrogels can deliver immune cells, immune checkpoint blockade (ICB) antibodies, and several other immunoregulatory factors [195]. In addition, they can be applied after surgery to eliminate the remaining cancer cells and prevent cancer cell recurrence. Based on these characteristics, Zhang and his coworkers developed a type of tumoricidal immunity hydrogel that contained a tumor-homing immune nanoregulator and chemotactic CXC chemokine ligand 10, which were used for indoleamine 2,3-dioxygenase-1 suppression, immunogenic cell death induction, and sustained infiltration of T-cells after GBM surgical resection [196]. 

Others fabricated thermo-reversible PEG-g-chitosan hydrogels (PCgels) for the local delivery and controlled release of T lymphocytes. Compared to the control Matrigel, T lymphocytes entrapped in the hydrogel showed better activity against glioblastoma cells, which may be due to the optimum pore size of the gel allowing efficient lymphocyte invasiveness [197]. The thermo-reversible hydrogel was also applied for a sustained and steady release of chemo- and immunotherapeutic agents to the brain after injection to the resection cavity of glioma. This in situ hydrogel delivery system consisted of a PLGA_1750_-PEG_1500_-PLGA_1750_ framework containing glioma homing peptide modified paclitaxel targeting NPs (PNP_PTX_) and mannitolated immunoadjuvant CpG-targeting NPs (MNP_CpG_). The formulation was in a solid state at room temperature and transformed to a gel at body temperature after injection into the resection cavity. Sustained release of PNP_PTX_ from the gel led to a targeting of the residual infiltration of glioma cells and the production of tumor antigens, whereas MNP_CpG_ targeted and activated antigen-presenting cells, which promoted tumor antigen presentation ability and activated CD8^+^ T and NK cells to counteract the immunosuppression of the glioma microenvironment [198] Of note, this delivery system is in its early stages, and only a few studies, mainly related to chemotherapy, have reported on it.

##### DNA-Based Nanocarriers

DNA nanocarriers are a new and attractive group of carriers that could be used to deliver various therapeutic agents [199,200]. For instance, a tetrahedral framework nucleic acid (tFNA) NP, a type of DNA-based nanocarrier, was designed for the delivery of TMZ to the GBM tumor, and it showed enhanced lethality in TMZ-sensitive cells and an attenuation of drug resistance in TMZ-resistant cells via decreasing the expression of O6-methylguanine-DNA-methyltransferase [201]. 

In addition to application in chemotherapy, DNA-based nanocarriers have extraordinary properties such as permeability, high structural programmability, high biocompatibility, high flexibility, and precise spatial addressability appropriate for carrying various immunotherapeutic agents. Thus, they can be applied to affect the receptor of unmethylated CpG oligonucleotides (Toll-like receptor 9 (TLR9)) and immune checkpoint receptors, and can be utilized as DNA vaccines [202]. CpG oligonucleotides conjugated to a tFNA in combination with siRNA was shown to enhance the immunotherapeutic efficacy on cancer cells (compared to monotherapy) via inducing the repolarization of M2 macrophages to an M1 phenotype [203].

##### Other Types of Nanomaterials 

In addition to the aforementioned materials, other types of nanomaterials have also been applied in GBM immunotherapy, including inorganic NPs such as Cu, Au, and TiO_2_ [204,205,206]. For instance, surface-functionalized CuSe NPs containing indoximod (IND, an inhibitor of indoleamine-2,3-dioxygenease at the tumor site) and JQ1 (inhibitor of PD-L1 expression in tumor cells) were synthesized. This was followed by covering the NP surface with tumor cell membrane to improve the targetability towards cancer tissues. The fabricated nanoformulation could reprogram the tumor immunosuppressive microenvironment and activate the immune response via increasing the number of M1-phenotype macrophages (by increasing ROS levels by near infrared II (NIR II) irradiation) and anti-tumor CD8^+^ T cells at the tumor site along with reducing PD-L1 expression and Treg cells infiltration [207].

Silica-based NPs are another group of nanomaterials used for GBM immunotherapy. The porous structure of these particles provides a good opportunity to load therapeutic compounds. Cyclic diguanylate monophosphate (cdGMP), an agonist of stimulator of interferon gene (STING), was encapsulated inside these pores to fabricate immunomodulatory NPs. This nanoformulation showed anti-tumor effects by enhancing the release of the proinflammatory cytokine IFN β, after uptake by APCs and the recruitment of macrophages and DCs [208]. Immune cells can be used as delivery agents as well. A theranostic NP was constructed using fluorescent polymer (BPLP)-polylactide copolymers (BPLP-PLAs) NPs containing doxorubicin (DOX). The fabricated NPs were attached to chimeric antigen receptor (CAR) T cells via a pH-responsive bond allowing the NPs to be released into the tumor tissue. Moreover, CAR-T cells were functionalized with targeted-quadruple-mutant of IL13 (TQM-13) that has affinity for the IL13-receptor-α2 (IL13Rα2), which is overexpressed on the surface of GBM cells [209].

The combination of radiotherapy and gold NPs coated with the outer membrane of *E. coli* has also been reported to be applicable in treating GBM cancer. NPs acted as immunomodulatory and radiosensitizing agents and induced tumor cell death, which was related to increased ROS levels, the production of TNF-α, and chemotaxis of macrophages [210]. In addition to their role as therapeutic carriers, nanomaterials can be implemented as imaging agents to monitor and trace the therapeutic process. For instance, iron oxide NPs served as labeling agents for different types of immune cells [211,212]. In addition, ultra-small iron oxide NPs (USPIOs) have been used to monitor the recruitment of monocytes to the GBM site after administration of amphotericin B. These NPs were taken up by cells of the reticuloendothelial system (such as monocytes) after being administrated to the blood. Amphotericin B induces the accumulation of pro-inflammatory monocytes in GBM tissue; labeling monocytes with USPIOs enables real-time monitoring of this process via magnetic resonance imaging (MRI) [213]. 

In other recent work, a complex of bovine serum albumin, manganese oxide, and salinomycin was used to fabricate an NP that could reprogram the tumor microenvironment. This was achieved via targeting the repolarization of macrophages from the M2 to the M1 phenotype, resulting from the presence of both salinomycin and Mn^2+^ at the tumor site. Moreover, Mn^2+^ provided the possibility of treatment monitoring using MRI [214]. 

Several other nanomaterials developed with the aim of implementation in GBM immunotherapy are summarized in Table 3. 

#### 4.2.2. Nanomaterials as Carrier for Virus Compounds

Nanomaterials could also act as vehicles for the delivery of viral compounds that have roles in the immunotherapy of cancer cells. In addition, it is a new field that could be used in future for producing a more effective method for glioblastoma immunotherapy benefiting from both viral effects and nanomedicine without concerning the contamination resulted from virus infection. Recently, the herpes simplex virus type I thymidine kinase (HSV-tk) gene was delivered via different types of nanocarriers into the glioma cancer cells to increase the effectiveness of chemotherapy [225,226]. Although these studies were about the combination of gene therapy and chemotherapy, they could be considered as a preface for future investigations on the delivery of virus compounds with nanomaterials. 

Besides the aforementioned examples, in some other cases, researchers have tried to prepare viral mimic nanocarriers with the ability to influence the immune system. For instance, Gao et al. fabricated a type of nanocarrier which contained a nucleic acid nanogel for reprogramming the macrophages and microglia for the treatment of glioblastoma. They used erythrocytes membranes functionalized with M2pep and HA2 peptides that coated a DNA nanogel containing miR155. The presence of the M2pep peptide could target the viral mimic nanoparticle towards M2-microglial and macrophage cells and HA2 peptide enhances the fusion of particles with endosomal membrane leading to release of DNA nanogel inside the cells. Inside the cells and in the presence of ribonuclease H (RNase H), miR155 compounds are released and reprogram the M2-phase cells into M1-type via the downregulation of the expression of anti-inflammatory proteins (such as suppressor of cytokine signaling 1 (SOCS-1)) in the cells, and the elevation of the expression of pro-inflammatory cytokines such as the inducible nitrogen synthase, IL-6, and TNF-α. These features led to the conversion of the immunosuppressive microenvironment of cancer tissue to an immunoactive one (Figure 9) [227].

### 4.3. Three-Dimensional Bioprinted Platforms and GBM Immunotherapy

Another exciting option for use in GBM immunotherapy is the application of 3 dimensional (3D) printed scaffolds. These scaffolds are commonly used as ideal models for monitoring treatment effects on targeted tissues without using animal models. Thus, 3D culturing models were introduced to overcome the deficiency of 2D cell culturing systems in which the microenvironment of cancer cells (and any other cell type) needs to be provided [228]. Among the most appealing 3D culturing methods is 3D bioprinting, which is based on fabricating a 3D structure of cancer tissue using computer design for the layer-by-layer composition of polymeric materials. Cells can be inserted into the tissue structure during the preparation process or after the printing process [228]. 

A type of 3D bio-ink platform was developed by creating a fibrin glioblastoma bio-ink consisting of patient-derived glioblastoma cells, microglia, and astrocytes. In addition, perfusable blood vessels were constructed using a sacrificial bio-ink coated with brain pericytes and endothelial cells. The effects of therapeutic agents in this 3D culturing platform were compared to in vivo and 2D culture settings. Similar results with regard to growth curves, drug response, and genetic signature were obtained for GBM cells grown in the 3D printed bio-ink platform and in vivo mouse cancer models, whereas considerable differences were observed compared to 2D culturing [229]. There are different technologies for the fabrication of 3D printed structures for pharmaceutical applications, such as stereolithography appearance (SLA), fused deposition modeling (FDM), and binder extrusion printing. Three-dimensional-printing could be a promising tool toward achieving personalized medicine via designing and developing patient-specific models with the ability to mimic individual responses to treatment [230]. This is a fast technique allowing for the fabrication of different tissues with 3D topologies that could generate reliable results by biomimicry of the tissue microenvironment. Furthermore, this approach can aid in evaluating the communication between cells and identifying suitable transport mechanisms of drugs and molecules [231]. 

Extrusion bioprinting is the standard bioprinting method used to fabricate 3D prints of GBM cancer; however, it has limitations that include subjecting the cells to shear stress during the extrusion process, and other bioprinting techniques are therefore under investigation. In the case of patient-specific models, researchers have used patient-derived cells to develop organ-on-a-chip models to evaluate “real” therapeutic responses. To assess the effect of the tumor microenvironment on tumor cell growth, a 3D printed scaffold containing hyaluronic acid hydrogel was fabricated, inside of which the development of glioblastoma cancer stem cells (with/without neural precursor cells and astrocytes) was assessed in the presence and absence of macrophages. This study showed that cancer cell invasiveness, proliferation, and drug resistance were enhanced in the presence of macrophages and astrocytes, confirming functional crosstalk between different cell types in the tumor microenvironment [232].

Three-dimensional bioprinting models also have the ability to assess GBM responses to immunotherapy. For instance, Heinrich et al. developed a mini-brain using two-step 3D-bioprinting to evaluate the crosstalk between GBM cells and macrophages as well as the effects of therapeutic agents on these interactions. They showed that the GBM cells recruit the macrophages and convert them into glioblastoma-associated macrophages (GAMs)-specific phenotype. These macrophages subsequently guided the propagation and metastasis of the GBM cells in the mini-brains. Evaluation of the effects of different therapeutic agents (carmustine (BCNU), AS1517499, and BLZ945) in this 3D model indicated that the IC_50_ values were much higher than those in the 2D tests, which was attributed to easier cell penetration of drugs in 2D models. Moreover, the presence of GAMs in the microenvironment of cancer cells could enhance the growth rate of cancer cells; however, these cells showed more sensitivity to the chemotherapeutic drugs and BCNU could significantly prevent the growth of cancer cells cocultured with macrophages (compared to the monocultured cells). Lastly, they showed that preventing the expression of some specific genes in macrophages cocultured with tumor cells (using the immunomodulators) could decrease the invasiveness of the tumor cells, confirming crosstalk between these macrophages and GBM cells (Figure 10) [233]. 

A microfluidics-based GBM-on-a-chip system has also been explored for the evaluation of the heterogeneity of immunosuppressive tumor microenvironments and optimization of immunotherapy targeted against the PD-1 checkpoint for different GBM subtypes. The results revealed that GBM-subtype-specific immune and epigenetic signatures could affect immunosuppressive mechanisms. Moreover, compared to proneural GBM, the mesenchymal GBM niche attracted high numbers of CD163^+^ TAMs and expressed elevated levels of PD-1/PD-L1 immune checkpoints and various cytokines (TGF-β1, IL-10, and CSF-1) [234].

## 5. Conclusions

In recent years, multiple strategies have been aimed at improving the efficacy of immunotherapy against GBM. Because of the poor prognosis and few effective treatment options, the potential for immunotherapy to benefit patients with GBM is of great interest. However, the BBB restricts the straight-forward delivery of immunotherapies, and the immunosuppressive GBM microenvironment is a challenge with regards to treating GBM with immune modulators. Developing efficient and clinically relevant immunotherapies such as OVs that express immunomodulators and immune checkpoint inhibitors should therefore be a prioritized subject of further research to improve patient outcomes. In addition, many issues related to CNS tumors should be addressed and considered as well; these include the heterogeneous nature of cancer and the presence of glioma stem cells. Rational combination therapies targeting multiple disease mechanisms/abnormalities within GBM tumors can induce a long-lasting anti-tumor immunity and may effectively overcome therapeutic resistance. While immunotherapy involving NPs has attracted significant attention for cancer treatment, the clinical translation of such immunostimulatory NPs remains a challenge. A primary consideration is the safety of immunostimulatory NPs, as they interact with the immune system and must be analyzed carefully. It is also important to determine whether NPs affect the intracellular signaling system when assessing their toxicity. Autoimmunity against NPs may occur when the immune system identifies them as foreign when interacting with serum proteins. NPs must be designed in such a way that they do not induce hypersensitivity, allergic sensitization, or ROS production and are rapidly cleared from the body. In general, multi-level targeted treatment for cancer, such as a mix of immunomodulators, viral vectors, and immunostimulatory NPs, is an exciting area of study. Although the clinical applicability of these novel (combined) therapies still has significant limits, their potential therapeutic efficacy cannot be ignored and warrants future research to overcome the remaining hurdles [220]. 

## Figures and Tables

**Figure 1 cancers-14-03698-f001:**
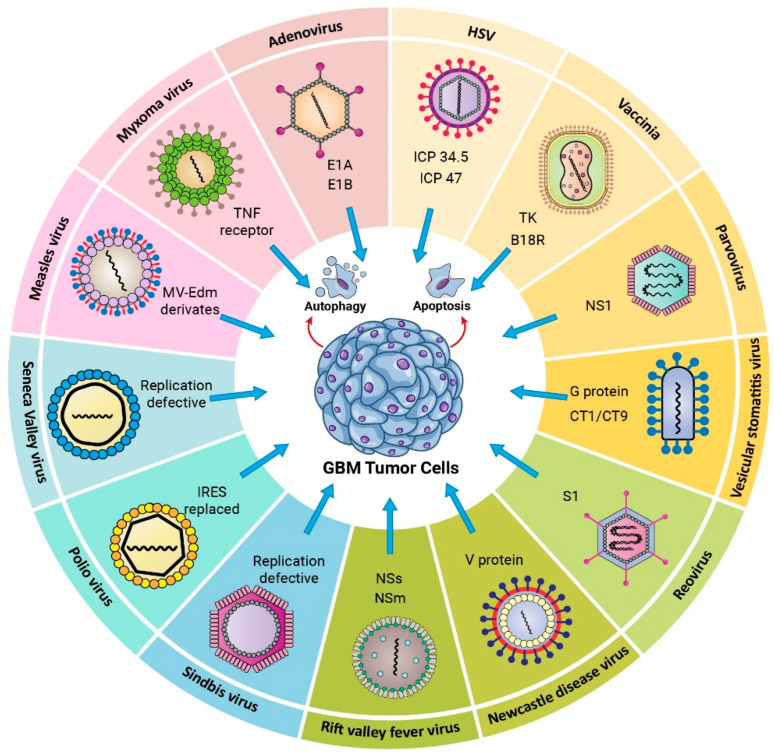
Oncolytic viruses used in Glioblastoma virotherapy. HSV (Herpes simplex virus); ICP34.5,ICP47 knocked out. Vaccinia virus; Tk, B18R knocked out. Parvovirus; NS1 knocked out. Vesicular Stomatitis Virus; M51,G protein, CT1/CT9 knocked out. Reovirus; SL knocked out. Newcastle disease virus; V protein knocked out. Rift Valley fever virus; NSs, NSm knocked out. Sindbis virus (Replication deficient). Poliovirus (IRES replaced), Seneca virus (Replication deficient), Measles (MV-Edem derivates), Myoxoma; TNF receptor knocked out. Adenovirus; E1A, E1B knocked out.

**Figure 3 cancers-14-03698-f003:**
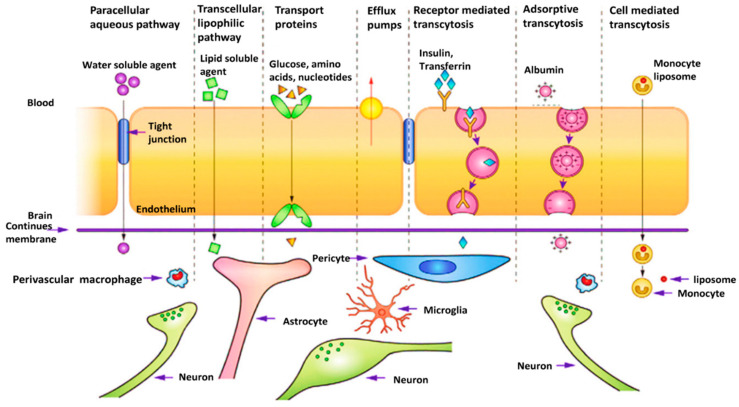
Different transportation mechanisms could be used for passing the BBB and delivering the drug to the glioblastoma cells. There are two main mechanisms for transporting molecules through the BBB: the paracellular pathway and the transcellular pathway which contains four different subparts: Receptor-mediated transcytosis, adsorptive-mediated, carrier/transporter-mediated transcytosis, and cell-mediated transcytosis. Reprinted with permission from [154].

**Figure 4 cancers-14-03698-f004:**
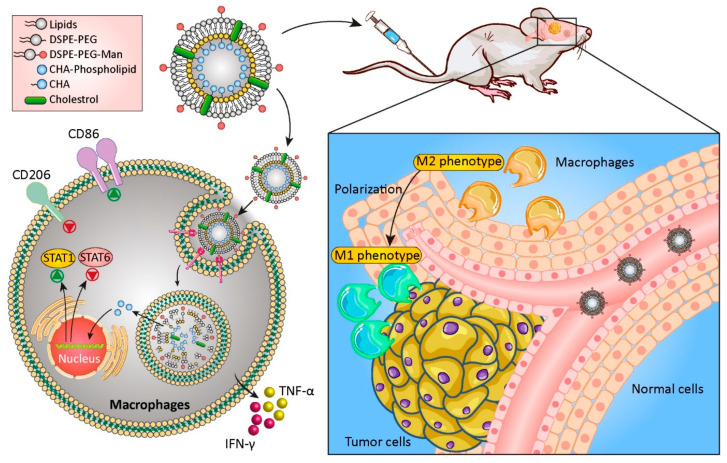
Schematic of applying chlorogenic acid-loaded mannosylated liposomes for immune targeting GBM therapy. Abbreviations: TMZ: Temozolomide, CHA: Chlorogenic acid, Lipo: Liposomes, DSPE-PEG: 1, 2-Distearoyl-sn-glycero-3-phosphoethanolamine-Poly(ethylene glycol), Man: Mannose, TNF-α: Tumor necrosis factor alpha, IFN-γ: Interferon gamma, STAT1: Signal Transducer And Activator Of Transcription 1. Reproduced from [176].

**Figure 5 cancers-14-03698-f005:**
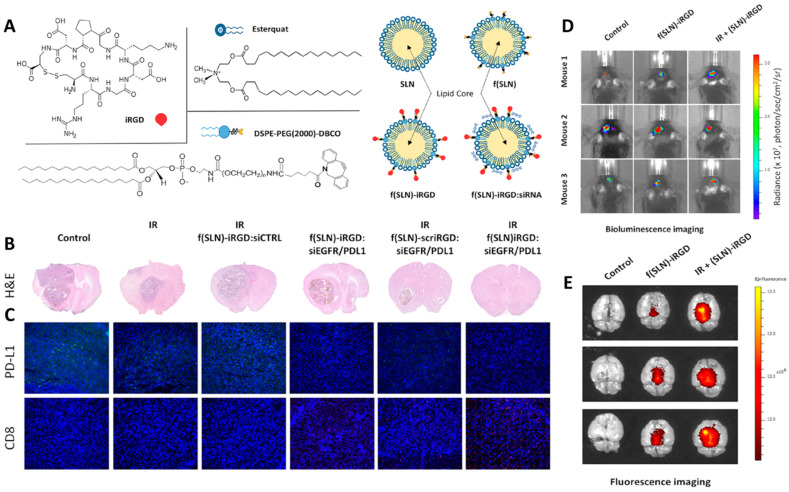
(**A**) Chemical structure and schematic of siRNA-loaded solid lipid NPs. (**B**) H and E images and (**C**) immunohistological analysis of brain sections after exposure to different treatments (anti-PD-L1 and anti-CD8 antibodies were used for immunohistochemistry). (**D**) Bioluminescence and (**E**) fluorescence images of the mouse brain treated with siRNA-loaded SLNs in the absence and presence of radiation therapy. Abbreviations: f(SLN): functionalized solid lipid nanoparticle, siCTRL: control scrambled siRNA, EGFR: Epidermal growth factor receptor, H and E: Hematoxylin and eosin, PDL1: Programmed death-ligand 1. Reprinted with permission from [181].

**Figure 6 cancers-14-03698-f006:**
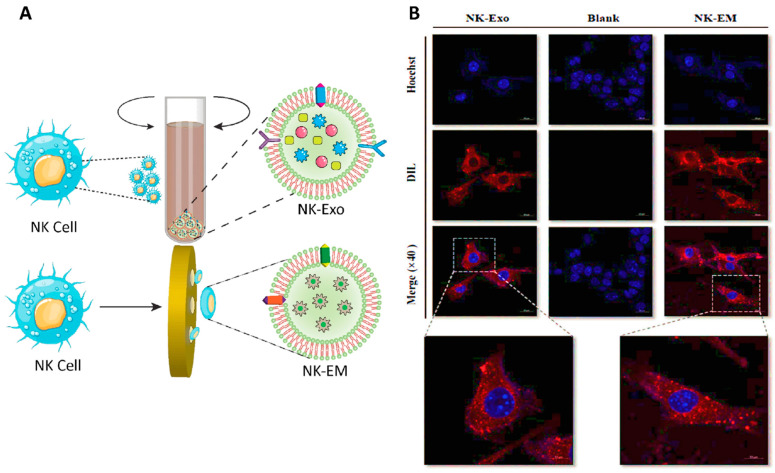
(**A**) Schematic of the extraction of exosome mimetics from NK cells (NK-EM). (**B**) Confocal images of D54/F cells treated with NK exosomes (NK-Exo) and NK-EM. Abbreviations: NK-Exo: Exosomes from natural killer cell, NK-EM: Exosome mimetics from natural killer cells, AKT: Protein kinase B, GADPH: Glyceraldehyde 3-phosphate dehydrogenase. Reprinted with permission from [188].

**Figure 7 cancers-14-03698-f007:**
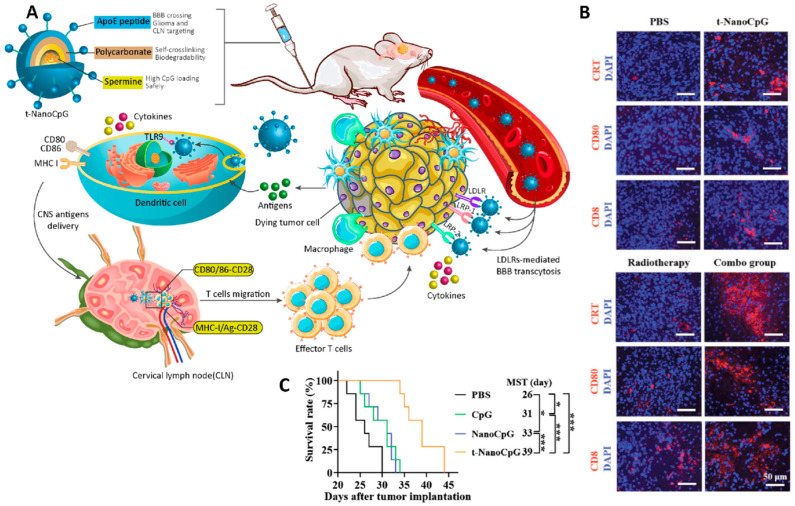
(**A**) Schematic representation of the intravenous brain delivery of a smartly targeted ApoE-functionalized polymersome CpG ODN nanoformulation for the immunotherapy treatment of GBM in mice. (**B**) The results of immunohistochemical staining of the tumor section of the brain were treated with different formulations. Combined immunotherapy and radiotherapy led to more ICD, as well as increased numbers of activated APCs and CD8^+^ T cells as compared to monotherapy. (**C**) Survival rate curve of mice treated with different formulations (* *p* < 0.05, and *** *p* < 0.001). Abbreviations: ApoE: Apolipoprotein E, IFN-γ: Interferon gamma, TNF-α: Tumor necrosis factor alpha, IL-6: Interleukin 6, CNS: Central nervous system, CLN: Cervical lymph node, PBS: Phosphate buffer saline. Reprinted with permission from [192].

**Figure 8 cancers-14-03698-f008:**
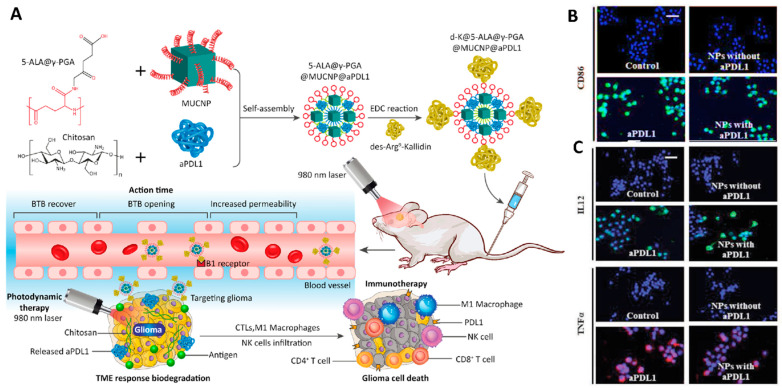
(**A**) Schematic images of the preparation and therapeutic application of multi-functional polymeric NPs developed for the photodynamic- and immunotherapeutic treatment of GBM. (**B**,**C**) Immunofluorescent staining confirmed the effect of NPs on the activation of tumor-associated macrophages. The expression levels of CD86 (C-up), major histocompatibility complex (MHC) II (C-down), interleukin 12 (IL12), and tumor necrosis factor-α (TNFα) changed after exposure to different formulations (scale bar = 30 µm). Abbreviations: PGA: Poly(γ-glutamic acid), MUCNP: Magnetic up conversion nanoparticles, PDL1: Programmed death-ligand 1, BTB: Blood tumor barrier, TME: Tumor micro environment, NK cells: Natural killer cells, IL12: Interleukin 12, TNFα: Tumor necrosis factor-α. Reprinted with permission from [194].

**Figure 9 cancers-14-03698-f009:**
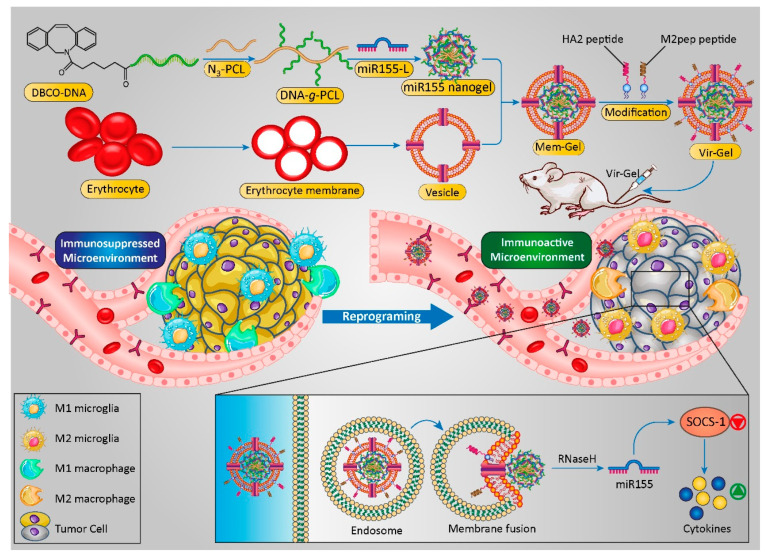
Scheme of virus-like nanoparticles used for GBM immunotherapy. This new type of vehicle was fabricated via the utilization of the erythrocyte’s membrane containing two types of peptides in the structure of the membrane and the entrapment of microRNA nanogel inside itself. The transmembrane peptides targeted the particles towards M2-phase macrophages and miRNA enhanced the polarization of macrophages to M1 phase. Reproduced from [227].

**Figure 10 cancers-14-03698-f010:**
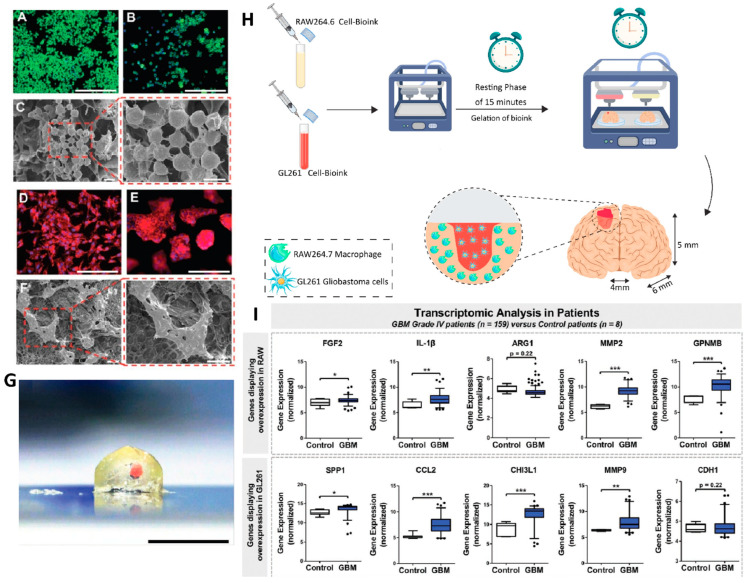
Fluorescence ((**A**,**B**), scale bar= 200 µm) and SEM ((**C**), scale bar= 10 µm) images of RAW264.7 macrophages (in a culture plate (**A**) and on top of the crosslinked construct (**B**)). Fluorescence ((**D**,**E**), Scale bar= 200 µm) and the SEM ((**F**), Scale bar= 10 µm) images of the attached GL261 glioblastoma cells (in a culture plate (**D**) and on top of the crosslinked construct (**E**)). (**G**) Cross-section of the fabricated mini-brain, in which the GBM tumor is highlighted in red (scale bar= 5 mm). (**H**) Schematic representation of the fabrication process and a cross-sectional and close-up view of the resulting mini-brain. (**I**) Results of transcriptomic analysis extracted from the public database GEO (GSE16011) for genes that are upregulated in RAW264.7 macrophages and GL261 cells (* *p* < 0.05, ** *p* < 0.01, and *** *p* < 0.001). Reprinted with permission from [233].

**Table 1 cancers-14-03698-t001:** DNA and RNA viruses proposed as Glioma oncolytic agents in clinical trial.

Virus Type	Virus Name	Modifications	Efficacy	Clinical Phase	Status	Trial Name (ClinicalTrial.gov)
HSV-1	G207	γ_1_34.5 loci deletion. ICP6 truncation.	Tumor cells are eliminated, necrosis occurs, and no toxin is produced.	I	Completed	NCT00028158
	rQNestin34.5	ICP6 deletion. γ_1_34.5 expression under nestin promoter.	Enhanced oncolytic activity in vitro and in an in vivo model.	I	Recruiting	NCT03152318
	HSV-1716	γ_1_34.5 loci partial deletion.	Infection and death of tumor cells.	I	Terminated	NCT02031965
	M032	2 copies of γ134.5 deletion. Human IL-12 expression.	Causes the tumor cell to synthesize and secrete the immunity-stimulating protein interleukin-12 (IL-12)	I	Active, not Recruiting	NCT02062827
	C134	γ_1_34.5 loci deletion. HCMV’s IRS1 protein expression.	Tumor volume reduction and improved surveillance. Improved replication and increased in vivo survival.	I	Recruiting	NCT03657576
Adeno virus	DNX-2401	24 base pair deletion in E1A gene. Engineered RGD-4 C binding motif.	Improved virus replication in cancer cells via Rb-pathway deficiency. Improved specificity by targeting tumor cells via αvβ3 and αvβ5.	II	Completed	NCT02798406
	DNX-2440	24 base pair deletion in E1A gene. Engineered RGD-4 C binding motif. OX40L insertion.	Improved virus replication in cancer cells via Rb-pathway-deficiency. Improved specificity by targeting tumor cells via αvβ3 and αvβ5. Stimulation of immune responses.	I	Recruiting	NCT03714334
Vaccinia virus	TG6002	TK deletion. ribonucleotide reductase genes deletion. FCU1 expression.	Survival in s.c. and i.c. over an extended period. Synergy with 5FC in an i.c. model.	I	Unknown	NCT03294486
Parvo virus	Parvo virus H-1 (ParvOryx)	Non-engineered virus.	Tumors in complete remission. Cathepsin B activation induces cell death in H-1PV with a complete remission of the tumors. Selective infection, no toxicity, in vivo reduction in tumor volume	II	Completed	NCT01301430
Reovirus	REOLYSIN	Non-engineered reovirus type 3.	Safe, well tolerated (no significant toxicity)	I	Completed	NCT00528684
	Polio/Rhinovirus Recombinant (PVS-RIPO)	PVS-RIPO consists of the genome of the live attenuated serotype 1 (Sabin) vaccine strain of poliovirus containing an internal ribosomal entry site (IRES) of human rhinovirus type-2.	Attenuation of neurovirulence. Attenuation determinants mapping to the coding regions for the type 1 (Sabin) capsid13 and RNA-dependent RNA polymerase.	I	Active, not Recruiting	NCT03043391
NDV WT	----------	Apoptosis induction. Tumor volume should be decreased. NDV causes idiopathic cardiomyopathy (ICD). The combination of TMZ has synergistic effects. Reduces the number of tumors and prolongs survival time.	Safe, well tolerated	I	Withdrawn	NCT01174537

**Table 2 cancers-14-03698-t002:** Clinical trials on the effectiveness of modified oncolytic viruses against gliomas.

Trial Name (ClinicalTrial.gov)	Phase	Virus Used/Mode of Action	Associated Treatments	Primary Endpoint	Results
NCT03636477	I	Ad-RTS-hIL-12	Veledimex + Nivolumab	Safety and feasibility	Active, not recruiting
NCT02062827	I	M032-HSV-1	-	Highest safe dose/MTD	Ongoing, Recruiting
NCT03330197	I/II	Ad-RTS-hIL-12	Veledimex	Safety and tolerability	Ongoing, Recruiting
NCT01811992	I	Ad-hCMV-TK and Ad-hCMV-Flt3L	-	Maximum Tolerated Dose (MTD)	Active, not recruiting
NCT03714334	I	DNX-2440	-	Incidence of Treatment-Emergent Adverse Events	Ongoing, Recruiting

**Table 3 cancers-14-03698-t003:** Application of nanomaterials for GBM immunotherapy.

Type of Nanomaterial	Therapeutic Compounds	Mechanism of Immunotherapy	Ref.
pH-sensitive polymeric micelles	- Anti-PD-1antibodies - Epirubicin	- Inducing immunogenic cell death - Eliminating the immunosuppressive myeloid-derived suppressor cells - Reducing the expression of PD-L1	[215]
Polycaprolactone (PCL):PEG:PCL polymer hydrogel	Anti-PD-1 antibody	Increasing IFN-γ and TNF-α levels	[216]
High-density lipoprotein (HDL)-mimicking nanodiscs	- CpG - Docetaxel	- Increasing the response of CD8^+^ T cell - Inducing cancer cell death and releasing the tumor antigens and damage-associated molecular pattern molecules (DAMPs) into the TME - Activation of macrophages and DCs	[217]
Smart redox responsive DOX loaded mesoporous silica nanoparticles	- DOX - Asp-Glu-Val-Asp (DEVD) peptide	- Activation of cytotoxic CD8^+^ T lymphocytes - Inhibition of CD4^+^ T cells - Upregulating the levels of antitumor cytokines	[218]
Polyglycerol-nanodiamond composites	- DOX	- Inducing autophagy to GBM cells - Improving the activation of DCs	[219]
Angiopep LipoPCB (TMZ+BAP/siTGF-β) smart nanoformulation	- TMZ - siRNA	- Down-regulating the expression of TGFβ - Enhancing the efficacy of FDA-approved drug TMZ - Regulating the proliferation of other T cells, including T effect cells (Teff), T regulation cells (Treg), and cytotoxic lymphocyte (CTL).	[220]
Poly (β-l-malic acid) (PMLA) nanoparticles functionalized with anti-transferrin receptor (a-TfR) mAb	- Cetuximab - siRNAs	- Reducing the expression of PDL-1 - Downregulating the expression of serine/threonine-protein kinase CK2 and the wild-type/mutated epidermal growth factor receptor (EGFR/EGFRvIII) - Suppressing cancer stem cell marker expression (such as c-Myc, CD133, and nestin)	[221]
Cationic liposome functionalized with a single chain antibody fragment recognizing the transferrin receptor (TfRscFv)	- Human wildtype TP53 (wtp53) plasmid - Anti- PD-1 antibody	- Inhibiting tumor growth - Inducing tumor cell apoptosis - Enhancing intratumoral T-cell infiltration	[222]
Chitosan nanoparticles	- siRNA targeting Gal-1 (siGal-1)	- Reducing myeloid suppressor cells and regulatory T cells - Increasing CD4^+^ and CD8^+^ T cells.	[223]
Hydrogels contain tumor-homing immune nanoregulator (THINR)	- Mitoxantrone - siRNA targeted indoleamine 2,3-dioxygenase-1 - Chemokine ligand 10 (CXCL10)	- Triggering immunogenic cell death - Inducing DC maturation -Suppressing the IDO1 - Enhancing the recruitment of activated T cells	[196]
Microglial membrane coated Fe_3_O_4_ nanoparticles connected with siRNA	siRNA against PD-L1	- Decreasing the expression of PD-L1 protein - Increasing the ratio of effector T cells and regulatory T cells - Inducing GBM cells ferroptosis - Inducing DC cell maturation - Increasing the polarization of M2-type microglia to M1-type	[224]

Abbreviations: a-TfR: anti-transferrin receptor, CTL: Cytotoxic lymphocyte, CXCL10: Chemokine ligand 10, DAMPs: Damage-associated molecular pattern molecules, DCs: Dendritic cells, DOX: Doxorubicin, EGFR: Epidermal growth factor receptor, FDA: United States food and drug administration, GBM: Glioblastoma multiforme, HDL: High-density lipoprotein, IDO1: Indoleamine 2,3-Dioxygenase 1, IFN-γ: Interferon gamma, PCL: Polycaprolactone, PD-L1: Programmed death-ligand 1, PEG: Polyethylene glycol, PMLA: Poly (β-l-malic acid), TfR: Transferrin receptor, Teff: T effect cells, TGF-β: Transforming growth factor beta, THINR: tumor-homing immune nanoregulator, TME: Tumor microenvironment, TMZ: Temozolomide, TNF-α: Tumor necrosis factor alpha, Treg: T regulation cells, wtp53: Wildtype TP53.

## Data Availability

Not applicable.

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
