# Peer review of "Immunology Meets Bioengineering: Improving the Effectiveness of Glioblastoma Immunotherapy"

_cancers, 2022, doi:10.3390/cancers14153698_

Round 1

Reviewer 1 Report

Include other references along with reference 1. The reference is getting limited as self-citation is made here for a related paper.  The authors have discussed in details the limitations in targeting GBM by immunotherapeutic oncolytic viruses and possible solutions.

In my opinion the way the figures from other publications have been re-used with permissions to create figures for this review is not the best method. The authors should summarize the results from the published articles and create their own figures with boxes, charts and cartoons. Authors should summarize information of panels C D and E in Fig 9 and created their own summary figure. Figure 7 should be replaced with creative cartoons from the authors. There are too many figures reused (with permission) from other articles. In my opinion this overuse of figures from other publications compromise the originality of this current review. The authors have discussed the findings of these research articles and can summarize the findings in text and using few schematic figures  of their own. In figure 8 authors should include data of results of combined immunotherapy on mouse survival (if this is available in the original article).

In the legends of Figure 6, 7 etc  author should cite the author-wise reference of the published data not just the number. They followed this pattern in Figure 10 legend. The citation format should be consistent in all the figures.

Author Response

Comments and Suggestions for Authors

Include other references along with reference 1. The reference is getting limited as self-citation is made here for a related paper.  The authors have discussed in details the limitations in targeting GBM by immunotherapeutic oncolytic viruses and possible solutions.

Answer: Thank you for the constructive comment. We have added more references  for the statement (Page 2, Line 54).

  • In my opinion, the way the figures from other publications have been re-used with permissions to create figures for this review is not the best method. The authors should summarize the results from the published articles and create their figures with boxes, charts and cartoons.
  • Answer: Thank you so much for your valuable comment; we have designed new figures and replaced some of the figures with our drawing ones
  • Authors should summarize information of panels C D and E in Fig 9 and created their own summary figure.
  • Answer: Thank you very much for your comment; the following sentences are added to the main text (Page 25, Lines 912-922):

“They checked the effect of presence anti-PD-L1 antibody on activation of the tumor-associated macrophages (MΦ) via evaluating the expression level of costimulatory molecules like CD86, interleukin 12 (IL-12), and tumor necrosis factor-α (TNFα). Immunofluorescence results showed that the expression of these components in cells treated with antibody alone and nanoformulation contained antibody was higher than control and nanoformulation without antibody, that confirmed the effectiveness of utilizing this antibody in activation of immune response. Accordingly, the combination use of photodynamic therapy and immunotherapy could be an effective method for treatment of GBM cancer via affecting both immune system and changing the microenvironment of cancer tissue”

  • Figure 7 should be replaced with creative cartoons from the authors.
  • Answer: Thank you so much for your valuable comment, we have replaced part A of this figure with our drawing ones.
  • There are too many figures reused (with permission) from other articles. In my opinion this overuse of figures from other publications compromise the originality of this current review. The authors have discussed the findings of these research articles and can summarize the findings in text and using few schematic figures of their own.
  • Answer: Thank you so much for your valuable comment, we have replaced most of figures with our drawing ones. Figure 4 and 10 are deleted and a new figure (figure 9) is added to the manuscript.
  • In figure 8 authors should include data of results of combined immunotherapy on mouse survival (if this is available in the original article).
  • Reply: Thank you so much for your comment, the survival rate curve is added to the figure (Figure 7 in the revised manuscript).
  • In the legends of Figure 6, 7 etc author should cite the author-wise reference of the published data not just the number. They followed this pattern in Figure 10 legend. The citation format should be consistent in all the figures.
  • Answer: Correct references are added to the legends.

Reviewer 2 Report

The manuscript entitled: "Immmunology Meets Bioengineering: Improving the Effectiveness of Glioblastoma Immunotherapy" is a narrative and critical revision of the litterature focused on biothecnological and pharmacological strategies  to target immune system evasion  for the treatment of glioma. The manuscript  is well organized and  includes an extensive  description of  the immunology of glioma  and the potential strategies able to improve the effects of immunomodulators  drugs  as well as the increase in their delivery. I think that such a manuscript is of sure interest for researcher that work in the field of glioma as well as immunotherapy.  .   

Author Response

The manuscript entitled: "Immmunology Meets Bioengineering: Improving the Effectiveness of Glioblastoma Immunotherapy" is a narrative and critical revision of the litterature focused on biothecnological and pharmacological strategies to target immune system evasion for the treatment of glioma. The manuscript is well organized and includes an extensive description of the immunology of glioma and the potential strategies able to improve the effects of immunomodulators drugs as well as the increase in their delivery. I think that such a manuscript is of sure interest for researcher that work in the field of glioma as well as immunotherapy.   

Answer: The authors appreciate the positive feedback of the respected reviewer.

Reviewer 3 Report

In this review, the authors wished to discuss current and future immunotherapy strategies for GBM, with a focus on nanomaterials.

I find the review too dispersed and disorganized. I recommend a major revision that includes re-focusing the article on nanomaterials and GBM only. The parts on ICI and OV are too lengthy, broadly described and with outdated references. For the nanomaterials part, the review needs to be shortened and restructured. The authors can include some information (ex. general characteristics of nanoparticles – paragraph 4.1.1) in a text box or a figure, and directly delve in the applications in GBM. It is important to detail the model (cell culture, which mouse model, clinical trial) but no need to show data figures.

Main comments:

1.     The opening paragraphs, i.e., the introduction p1, and p2.1 and 2.2 on ICI, are too long, broad and do not discuss recent literature. They can be removed or summarized in a succinct preface or figure.

2.     Paragraphs 2.3.1 and 2.3.2 deal with IFNg-mediating cancer resistance to apoptosis and upregulation of immune checkpoint signaling. These paragraphs are not well placed in the review, as they come without any prior introduction of the immune context of GBM and focus narrowly on only this one cytokine pathway. In addition, they are not focused on GBM specifically, as the authors cite all types of cancers in which this signaling pathway has been studied.

3.     P2.3.3 “Immune-microenvironment in the CNS and local delivery of immunotherapeutics” should be the initial paragraph of the review article. However, unfortunately, its title does not reflect the paragraph content, as the authors do not fully describe the immune microenvironment of GBM! Furthermore, they lightly discuss GBM cancer stem cells (CSC) being at the basis of resistance to therapy, with outdated references. Taken the massive progress in the field, with the use of single cell technologies characterizing the diversity of immune cells in GBM and the intra-tumoral heterogeneity related to CSC profiles, this paragraph inappropriately reviews the current literature.

4.     Paragraph 2.3.4 dealing with ‘effector immune cells reaching GBM’ again is inappropriate. Half of the paragraph discusses mesenchymal stem cells, which are not ‘immune effector cells’. The second half brushes lightly on CAR-T and macrophages. This aspect needs to be significantly expanded with recent pre-clinical studies and clinical trials using these approaches. The authors need to specify the CAR-T targets and the strategies to modulate macrophages in GBM.

5.     Paragraph 2.3.5 – the immunosuppressive environment of GBM. The authors include inaccurate statements like “its microenvironment is associated with high inhibitory checkpoint expression” without supporting references. And only one approach to overcome immunosuppression is mentioned.

6.     P2.3.6, back to ICI! All ICI should be consolidated in one paragraph. Such a paragraph should first summarize the state of the field, i.e., first discuss phase III trials with anti-PD-1 in GBM that did not achieve beneficial outcomes.

7.     Paragraph 3.1. is a long list of OV that can be presented in a supplemental table. Instead, the focus should be placed on expanding 3.2. and the information in Table 9 and Figure 2.

Author Response

Reviewer #2

Comments and Suggestions for Authors

The manuscript entitled: "Immmunology Meets Bioengineering: Improving the Effectiveness of Glioblastoma Immunotherapy" is a narrative and critical revision of the litterature focused on biothecnological and pharmacological strategies to target immune system evasion for the treatment of glioma. The manuscript is well organized and includes an extensive description of the immunology of glioma and the potential strategies able to improve the effects of immunomodulators drugs as well as the increase in their delivery. I think that such a manuscript is of sure interest for researcher that work in the field of glioma as well as immunotherapy.   

Answer: The authors appreciate the positive feedback of the respected reviewer.

Comments and Suggestions for Authors

In this review, the authors wished to discuss current and future immunotherapy strategies for GBM, with a focus on nanomaterials.

I find the review too dispersed and disorganized. I recommend a major revision that includes re-focusing the article on nanomaterials and GBM only. The parts on ICI and OV are too lengthy, broadly described and with outdated references. For the nanomaterials part, the review needs to be shortened and restructured. The authors can include some information (ex. general characteristics of nanoparticles – paragraph 4.1.1) in a text box or a figure, and directly delve in the applications in GBM. It is important to detail the model (cell culture, which mouse model, clinical trial) but no need to show data figures.

Answer: Thank you so much for your valuable comment; we removed the general information about nanomaterials. Moreover, we redraw most of the figures. Some new sections are also added to the manuscript as follows:

"4.1. Engineering and Biology: Two pairs of eyes are better than one

Engineering is not just about the engine. In debates about engineering speak in biology, a common picture of what engineering is and how engineers operate recurs. A well-informed engineer at a drawing board creates designs for a polished and optimized product that "merely" has to be created in this picture of engineered artifacts, which are mostly mechanical or electrical gadgets. While this picture may capture some instances of engineering, it does not adequately represent the field's depth and breadth. It's important to note that it omits important aspects of engineering relevant to biology. A deliberate, goal-directed designer does not create biological systems as he/she does in engineering. However, this does not imply that the processes of evolution and human design are diametrically opposed. When we look at the design processes used by certain human engineers and innovators, we see a lot of parallels to evolution. Trial and error play an important part in the work of even the most imaginative innovators, for example. Human engineers, on the other hand, are more goal-oriented in the near term than nature, but their predictions about what a new invention may be used for often fall flat. As a result of their idealistic ideas of engineering, critics of the biology-engineering nexus typically disregard the iterative and error-prone process that really takes place[147].            One of the most severe kinds of cancer, malignant brain tumors have low survival rates, which haven't altered in the previous 60 years. In part, this is due to the particular anatomical, physiological, and immunological barriers that the brain presents. Innovative engineering solutions may be facilitated by the unique interaction of these obstacles. Cancer immunotherapy, which uses the body's own immune system to fight cancer, is becoming a common treatment for a wide range of tumors. Anatomical, physiological, and immunological challenges arise when working with the brain, which is a crucial organ. There are inherent technical problems in creating medicines that must function inside the brain and brain tumor immune milieu because of these specific anatomical and physiological restrictions. We may be able to use a wider range of technologies to develop personalized therapy techniques that can meet the specific biological restrictions of treating inside the brain, thereby allowing improved brain cancer immunotherapies[148]."

"4.2.2. Nanomaterials as carrier for virus compounds

Nanomaterials could also act as a vehicle for delivery of viral compounds that have roles in immunotherapy of cancer cells. In addition, it is a new field that could be used in future for producing a more effective method for glioblastoma immunotherapy benefiting from both viral effects and nanomedicine without concerning the contamination resulted from virus infection. Recently, the herpes simplex virus type I thymidine kinase (HSV-tk) gene was delivered via different types of nanocarriers into the glioma cancer cells to increase the effectiveness of chemotherapy[225,226]. Although these researches were about the combination of gene therapy and chemotherapy, they could be considered as a preface for future investigations on the delivery of virus compounds with nanomaterials.

Besides the aforementioned examples, in some other cases, researchers try to prepare viral mimic nanocarriers with the ability of influencing the immune system. For instance, Gao et al., fabricated a type of nanocarrier contained nucleic acid nanogel for reprogramming the macrophages and microglia for treatment of glioblastoma. They have used erythrocytes membranes functionalized with M2pep and HA2 peptides that coated a DNA nanogel containing miR155. The presence of M2pep peptide could target the viral mimic nanoparticle towards M2-microglial and macrophage cells and HA2 peptide enhances the fusion of particles with endosomal membrane leading to release of DNA nanogel inside the cells. Inside the cells and in the presence of ribonuclease H (RNase H), miR155 compounds are released and reprogram the M2-phase cells into M1-type via downregulating the expression of anti-inflammatory proteins (like suppressor of cytokine signaling 1 (SOCS-1)) in the cells, and elevating expression of pro-inflammatory cytokines such as the inducible nitrogen synthase, IL-6, and TNF-α. These features led to converting the immunosuppressive microenvironment of cancer tissue to an immunoactive one (Figure 9)[227].

Figure 9. Scheme of virus like nanoparticles used for GBM immunotherapy. This new type of vehicle was fabricated via utilizing erythrocytes membrane contained two types of peptides in the structure of membrane and entrapped microRNA nanogel inside itself. The transmembrane peptides targeted the particles towards M2-phase macrophages and miRNA enhanced the polarization of macrophages to M1 phase. Reproduced from [227]."

Main comments:

  1. The opening paragraphs, i.e., the introduction p1, and p2.1 and 2.2 on ICI, are too long, broad and do not discuss recent literature. They can be removed or summarized in a succinct preface or figure.

Answer: The introduction (page 1) was summarized. Paragraph 2, line 67 and 68 and lines 70-74 and reference number 6 were deleted.

Section number 2. Immune checkpoint inhibitors in cancer immunotherapy, was deleted.

Section 2.1, lines 119-127 and 129-139, 146-154 were deleted.

Section 2.2, lines 156-157, 168-170 and 173-174 were deleted.

  1. Paragraphs 2.3.1 and 2.3.2 deal with IFNg-mediating cancer resistance to apoptosis and upregulation of immune checkpoint signaling. These paragraphs are not well placed in the review, as they come without any prior introduction of the immune context of GBM and focus narrowly on only this one cytokine pathway. In addition, they are not focused on GBM specifically, as the authors cite all types of cancers in which this signaling pathway has been studied.

Answer: The paragraphs 2.3.1 and 2.3.2 were deleted.

  1. 3.3 “Immune-microenvironment in the CNS and local delivery of immunotherapeutics” should be the initial paragraph of the review article. However, unfortunately, its title does not reflect the paragraph content, as the authors do not fully describe the immune microenvironment of GBM! Furthermore, they lightly discuss GBM cancer stem cells (CSC) being at the basis of resistance to therapy, with outdated references. Taken the massive progress in the field, with the use of single cell technologies characterizing the diversity of immune cells in GBM and the intra-tumoral heterogeneity related to CSC profiles, this paragraph inappropriately reviews the current literature.

Answer: Paragraph 2.3.3 were edited completely. Lines 247-250 were deleted. Five new paragraphs were added to describe the immune microenvironment of GBM with new and Up to Date references (27-36, 39-50, 53-56 and 58-60).

  1. Paragraph 2.3.4 dealing with ‘effector immune cells reaching GBM’ again is inappropriate. Half of the paragraph discusses mesenchymal stem cells, which are not ‘immune effector cells. The second half brushes lightly on CAR-T and macrophages. This aspect needs to be significantly expanded with recent pre-clinical studies and clinical trials using these approaches. The authors need to specify the CAR-T targets and the strategies to modulate macrophages in GBM.

Answer: The section was edited. Lines 267-285 were deleted. Three new paragraphs were added with Up to Date references.

  1. Paragraph 2.3.5 – the immunosuppressive environment of GBM. The authors include inaccurate statements like “its microenvironment is associated with high inhibitory checkpoint expression” without supporting references. And only one approach to overcome immunosuppression is mentioned.

Answer: Some parts of the section were added to ICI parts. Immunosuppressive nature of GBM was written based on cytokines and chemokines signaling in GBM tumors in recent clinical trials.

  1. 3.6, back to ICI! All ICI should be consolidated in one paragraph. Such a paragraph should first summarize the state of the field, i.e., first discuss phase III trials with anti-PD-1 in GBM that did not achieve beneficial outcomes.

Answer: The paragraph was edited and added to ICI parts as new p2.3. CD47 and CD137.

  1. Paragraph 3.1. is a long list of OV that can be presented in a supplemental table. Instead, the focus should be placed on expanding 3.2. and the information in Table 9 and Figure 2.

Answer: The Oncolytic viruses section was summarized and 8 tables of the section merged in one table as table number 1.

Round 2

Reviewer 1 Report

Revisions incorporated are appropriate. The manuscript is acceptable for publication.